# Independent phase modulation for quadruplex polarization channels enabled by chirality-assisted geometric-phase metasurfaces

Yueyi Yuan[1], Kuang Zhang [1✉], Badreddine Ratni[2], Qinghua Song [3], Xumin Ding [1,4✉], Qun Wu[1], Shah Nawaz Burokur [2✉] & Patrice Genevet[3✉]

Geometric-phase metasurfaces, recently utilized for controlling wavefronts of circular polarized (CP) electromagnetic waves, are drastically limited to the cross-polarization modality. Combining geometric with propagation phase allows to further control the co-polarized output channel, nevertheless addressing only similar functionality on both co-polarized outputs for the two different CP incident beams. Here we introduce the concept of chirality-assisted phase as a degree of freedom, which could decouple the two co-polarized outputs, and thus be an alternative solution for designing arbitrary modulated-phase meta-surfaces with distinct wavefront manipulation in all four CP output channels. Two meta-surfaces are demonstrated with four arbitrary refraction wavefronts, and orbital angular momentum modes with four independent topological charge, showcasing complete and independent manipulation of all possible CP channels in transmission. This additional phase addressing mechanism will lead to new components, ranging from broadband achromatic devices to the multiplexing of wavefronts for application in reconfigurable-beam antenna and wireless communication systems.

[1] Department of Microwave Engineering, Harbin Institute of Technology, 150001 Harbin, China. [2] LEME, UPL, Univ Paris Nanterre, 92410 Ville d'Avray, France. [3] Université Côte d'Azur, CNRS, Centre de Recherche sur l'Hétéro-Epitaxie et ses Applications (CRHEA), 06560 Valbonne, France. [4] Advanced Microscopy and Instrumentation Research Center, Harbin Institute of Technology, 150080 Harbin, China. ✉email: zhangkuang@hit.edu.cn; xuminding@hit.edu.cn; sburokur@parisnanterre.fr; pg@crhea.cnrs.fr

Artificially engineered metasurfaces have attracted great interests during the last two decades[1–5]. Due to the unique abilities to provide a large degree of control over the amplitude, phase, and polarization of local fields, metasurfaces find various applications in frequency regimes extending from microwaves to visible domains[6,7]. In particular, phase-gradient metasurfaces have emerged as a versatile platform for wave front engineering[8]. Compared to traditional bulky optical lenses relying on phase accumulation along propagation, metasurfaces can attain abrupt phase change within subwavelength thickness by suitably engineered inclusions that can enhance local interaction with waves, and thus have been explored to construct low-profile lenses[9–12], holograms[13–19], beam deflectors[20–22], spin-orbital angular momentum manipulation[23–28], information processing, and analog computations[29,30], among many others.

Since proposed by Pancharatnam and Berry[31,32], geometric phase, namely Pancharatnam–Berry (PB) phase, has been adopted in the manipulation of circularly polarized (CP) waves[33–39]. Especially on the platform of metasurfaces, PB phase can be conveniently introduced into transmitted (or reflected) waves by gradually varying local orientation from unit to unit. Ideally, there should be four channels by switching the handedness of CP input and output beams for PB phase based metasurfaces operating in transmission manner, including two cross-polarized channels (L–R, right-handed circular polarization (RHCP) output under left-handed circular polarization (LHCP) input, and R–L, LHCP output under RHCP input) and two co-polarized channels (L–L, LHCP output under LHCP input, and R–R, RHCP output under RHCP input). The inherent symmetrical response of PB phase results in that the functionality exhibited by metasurfaces in L–R channel and R–L channel would be equal and opposite. For instance, a geometric phase converging metalens for LHCP incidence will act as a diverging metalens for RHCP incidence. More recently, the combination of propagation and geometric phases is proposed to independently modulate the two cross-polarized output channels (L–R and R–L) under the orthogonal CP incidence states[40–44], but it has to be noted that the co-polarized output field are still left behind. The propagation phase along the fast and slow axis could affect the phase of co-polarized output fields, but it owns no selectivity for the handedness, which would result in the similar response for L–L and R–R output channels[45]. Obviously, a third phase resource is needed for the full phase modulation of CP beam. A direct property that can be utilized is the chirality, which can effectively generate circular dichroism with the illumination of orthogonal CP incidences[46,47]. It is demonstrated that chirality can be used to decouple the coherence between linearly polarized phase and amplitude response of metasurface to generate finite-energy airy beam[48]. Nevertheless, there still lacks a methodology that could integrate chirality into the phase modulation scheme as an additional degree of freedom to discriminate the wavefront of co-polarized L–L and R–R channels.

In this work, a general phase modulation scheme for the construction of metasurfaces enabling distinct wavefronts in all four CP channels is established. Chirality-assisted phase response, instead of amplitude response, namely circular dichroism, is proposed and then synthesized with propagation phase and PB phase to activate and distinguish wavefront engineering in all four co-polarized and cross-polarized channels under LHCP and RHCP wave illuminations. Accordingly, two metasurfaces are proposed and experimentally verified with different functionalities that in one case, beam deflection with arbitrary designer four refraction angles are obtained. With incident polarization state following the trace from RHCP to LHCP on the Poincaré sphere, evolution of diffraction order further confirms that all four CP channels are fully used. In another case, orbital angular

momentum (OAM) modes with four independent topological charges are proposed. This work provides an additional step on the prospective way towards multifunctional metasurface devices and can be possibly extended to other frequency regimes including visible wavelength.

## Results

**Principle of phase manipulation of full CP channels.** Here, we propose a general formalism for complete and separate phase manipulation of all four CP channels, where arbitrary and independent wavefronts can be achieved by altering the CP states of input and output ends, as schematically illustrated in Fig. 1. Taking generation of OAM modes as a demonstration example, four OAM mode with topological charge $l = 0$, 1, 2, and 3 are realized through different CP channels. The corresponding theoretical spatial phase distributions of the different output OAM modes and the schematic CP conversion process from input to output waves are exhibited in the insets of Fig. 1.

Different from previous works on birefringent metasurfaces[41,45], the aim of this work is to perform distinct phase-modulation in four CP transmission channels, which can be described by the four element transmission coefficients in the Jones matrix $T_{cir} = \begin{bmatrix} t_{LL} & t_{LR} \\ t_{RL} & t_{RR} \end{bmatrix}$. The equivalent metasurface system is supposed to be passive, lossless, matched, and reciprocal. Therefore, the four CP transmission coefficients (the first/second subscript represents the input/output CP state, L/R denotes the LHCP/RHCP state) with linear base can be described as follows:

$$t_{LL} = \frac{1}{2}\left[(t_{xx} + t_{yy}) + i \cdot (t_{xy} - t_{yx})\right], \tag{1a}$$

$$t_{LR} = \frac{1}{2}\left[(t_{xx} - t_{yy}) - i \cdot (t_{xy} + t_{yx})\right] \cdot e^{i \cdot 2\theta}, \tag{1b}$$

$$t_{RL} = \frac{1}{2}\left[(t_{xx} - t_{yy}) + i \cdot (t_{xy} + t_{yx})\right] \cdot e^{-i \cdot 2\theta}, \tag{1c}$$

$$t_{RR} = \frac{1}{2}\left[(t_{xx} + t_{yy}) - i \cdot (t_{xy} - t_{yx})\right], \tag{1d}$$

where $t_{xx} = |t_{xx}| \cdot e^{i \cdot \varphi_{xx}}$ and $t_{yy} = |t_{yy}| \cdot e^{i \cdot \varphi_{yy}}$ are the diagonal linear transmission coefficients, and $t_{xy} = |t_{xy}| \cdot e^{i \cdot \varphi_{xy}}$ and $t_{yx} = |t_{yx}| \cdot e^{i \cdot \varphi_{yx}}$ are the off-diagonal linear transmission coefficients. $\theta$ is the exterior rotation angle introduced by rotation matrix $M(\theta) = \begin{bmatrix} \cos\theta & \sin\theta \\ -\sin\theta & \cos\theta \end{bmatrix}$. Here, $t_{LL}$ and $t_{RR}$ are defined as co-polarized transmission channels, which maintain the polarization state of input waves. $t_{LR}$ and $t_{RL}$ represent cross-polarized channels, which flip the output fields into opposite CP state. The first components with totally same expression in $t_{LL}$ and $t_{RR}$ (or $t_{LR}$ and $t_{RL}$) can be labeled as $t_{propa}^{co} = |t_{propa}^{co}|e^{i \cdot \varphi_{propa}^{co}} = \frac{1}{2}(t_{xx} + t_{yy})$ ($t_{propa}^{cross} = |t_{propa}^{cross}|e^{i \cdot \varphi_{propa}^{cross}} = \frac{1}{2}(t_{xx} - t_{yy})$). The phase pattern of the two transmission components $\varphi_{propa}^{co}$ and $\varphi_{propa}^{cross}$ are uniquely dependent to propagation phase modulation, which would produce initial influence in both co-polarized and cross-polarized fields regardless of the incident CP state. Meanwhile, the second components carrying opposite symbols in $t_{LL}$ and $t_{RR}$ (or $t_{LR}$ and $t_{RL}$) can be extracted as $t_{chiral}^{co} = |t_{chiral}^{co}|e^{i \cdot \varphi_{chiral}^{co}} = \frac{1}{2} \cdot i \cdot (t_{xy} - t_{yx})$ ($t_{chiral}^{cross} = |t_{chiral}^{cross}|e^{i \cdot \varphi_{chiral}^{cross}} = \frac{1}{2} \cdot i \cdot (t_{xy} + t_{yx})$). The phase pattern of these two components $\varphi_{chiral}^{co}$ and $\varphi_{chiral}^{cross}$ are determined by chirality-assisted phase, indicating that the two components would be an additional degree of freedom to decouple inherent consistency between co-polarized channels. Moreover, the PB phase pattern

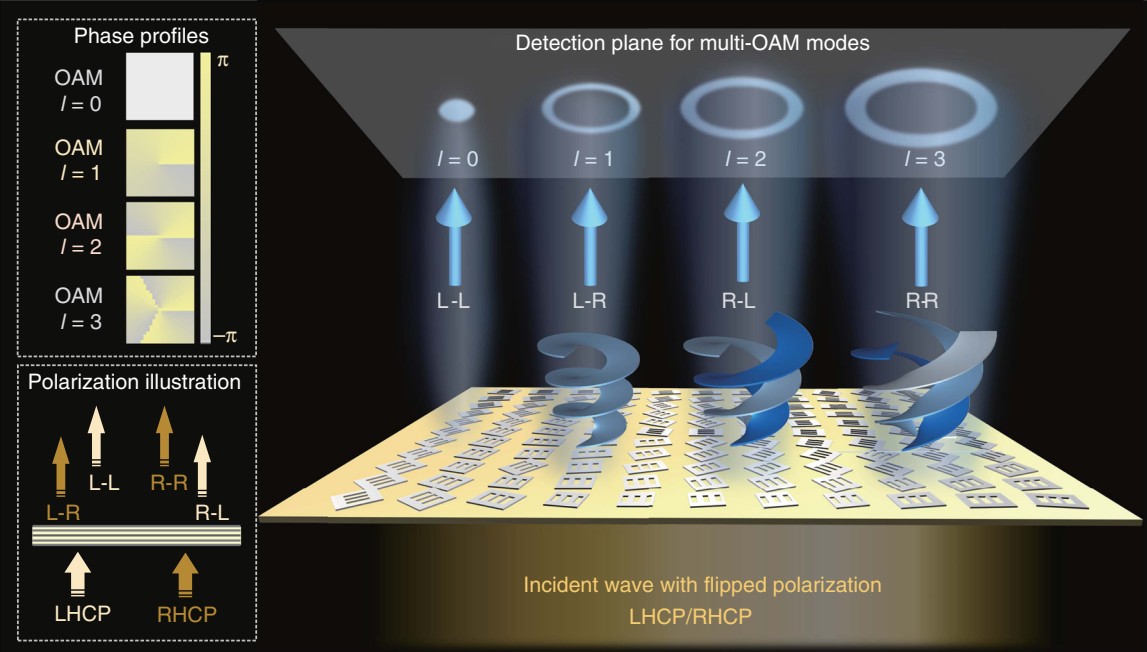

**Fig. 1 Schematic principle of proposed metasurface device for complete phase-modulation in quadruplex polarization channels.** Four vortex beams carrying OAM modes with $l = 0, 1, 2, 3$ are generated through L–L, L–R, R–L, R–R channels, respectively. The insets show the phase profiles of required OAM modes and the illustration of CP channels.

generated by rotating matrix $\varphi_{geo} = 2\theta$, which is sensitive to the incidence polarization state, influence only the two cross-polarized fields. Here, the plus or minus (±) sign represents the conjugate responses to LHCP or RHCP input states. According to Eq. (1a–d), it can be concluded that the full phase-modulation scheme is based on three phases that constitute three degrees of freedom to decouple the inherent consistencies between CP transmission coefficients: (i) propagation phase modulation is applied to define the initial phase profiles of two diagonal and two off-diagonal transmission elements $t_{LL}$ ($=t_{RR}$) and $t_{LR}$ ($=t_{RL}$) when there is no chirality-assisted and geometric phase responses, (ii) the chirality-assisted phase modulation is introduced to decouple the consistency between $t_{LL}$ and $t_{RR}$ while keeping $t_{LR}$ and $t_{RL}$ unchanged, and (iii) the geometric phase modulation can further realize distinct profiles of $t_{LR}$ and $t_{RL}$, meanwhile producing no effect on $t_{LL}$ and $t_{RR}$.

When the proposed metasurface is illuminated by LHCP incident wave $|\overrightarrow{L}\rangle = \begin{bmatrix} 1 \\ i \end{bmatrix}$ and RHCP incident wave $|\overrightarrow{R}\rangle = \begin{bmatrix} 1 \\ -i \end{bmatrix}$ respectively, the output electric fields can be expressed as:

$$\overrightarrow{E}_{out}^{L,in} = T \cdot |L\rangle$$
$$= \left(t_{propa}^{co} + t_{chiral}^{co}\right) \cdot |L\rangle + \left(t_{propa}^{cross} - t_{chiral}^{cross}\right) \cdot e^{i \cdot \varphi_{geo}} \cdot |R\rangle$$
$$= \overrightarrow{E}_{LL} + \overrightarrow{E}_{LR}$$

$$(2a)$$

$$\overrightarrow{E}_{out}^{R,in} = T \cdot |R\rangle$$
$$= \left(t_{propa}^{co} - t_{chiral}^{co}\right) \cdot |R\rangle + \left(t_{propa}^{cross} + t_{chiral}^{cross}\right) \cdot e^{-i \cdot \varphi_{geo}} \cdot |L\rangle$$
$$= \overrightarrow{E}_{RR} + \overrightarrow{E}_{RL}$$

$$(2b)$$

It can be observed from Eq. (2a, b) that under the orthogonal CP illuminations, the metasurface can produce four different output fields, including two co-polarized output components $\overrightarrow{E}_{LL}$

and $\overrightarrow{E}_{RR}$, and two cross-polarized components $\overrightarrow{E}_{LR}$ and $\overrightarrow{E}_{RL}$. Based on the analyses above, four CP transmission coefficients can be independently phase-modulated, where the inherent consistencies between four output phase patterns $\varphi_{LL}$, $\varphi_{LR}$, $\varphi_{RL}$, and $\varphi_{RR}$ are independently decoupled by propagation phase, chirality-assisted phase and geometric phase, respectively.

**Meta-atom design and verification.** Here a passive, reciprocal transmission-type meta-atom is applied to implement all these three phase modulation schemes, namely chirality-assisted phase, propagation phase, and PB phase. The proposed meta-atom is composed of five metallic layers and four dielectric substrates for realizing a 4th-order LC band-pass filter, whose geometric structure of meta-atom is introduced and schematically shown in Fig. 2a and the detailed operating mechanism is explained in Supplementary Note 1, Supplementary Figs. 1 and 2. In our designed meta-atom, propagation phase is provided by tailoring the width $p_x$ and length $p_y$ of the rectangular patch layers as shown in Fig. 2b. Figure 2c exhibits the phase profiles of all four outputs against $p_x$, where the phase response of the two co-polarized output channels show the same tendency $\varphi_{LL} = \varphi_{RR}$, as well as the two cross-polarized output phase delays $\varphi_{LR} = \varphi_{RL}$, while the co-polarized and cross-polarized phase responses exhibit differently $\varphi_{LL} \neq \varphi_{LR}$ ($\varphi_{RR} \neq \varphi_{RL}$) (the co-polarized and cross-polarized phase profile tendencies against both $p_x$ and $p_y$ are shown in Supplementary Fig. 3). That is to say, propagation phase can achieve independent manipulation of co-polarized and cross-polarized phase delays but does not show any selectivity on the spin state of incidence. In order to conquer this limit, chirality-assisted phase is introduced by the relative rotation of three similar patch layers of meta-atom with different angles, regarded as an interior rotation. Here orientation of middle layer is fixed with no further rotation, while the upper and lower layers are rotated with opposite angles $\vartheta$ respectively, as shown in Fig. 2d. A representative meta-atom selected with specific dimensions and different interior angle $\vartheta$ is simulated and presented in Fig. 2e. It can be seen that the two co-polarized phase profiles exhibit

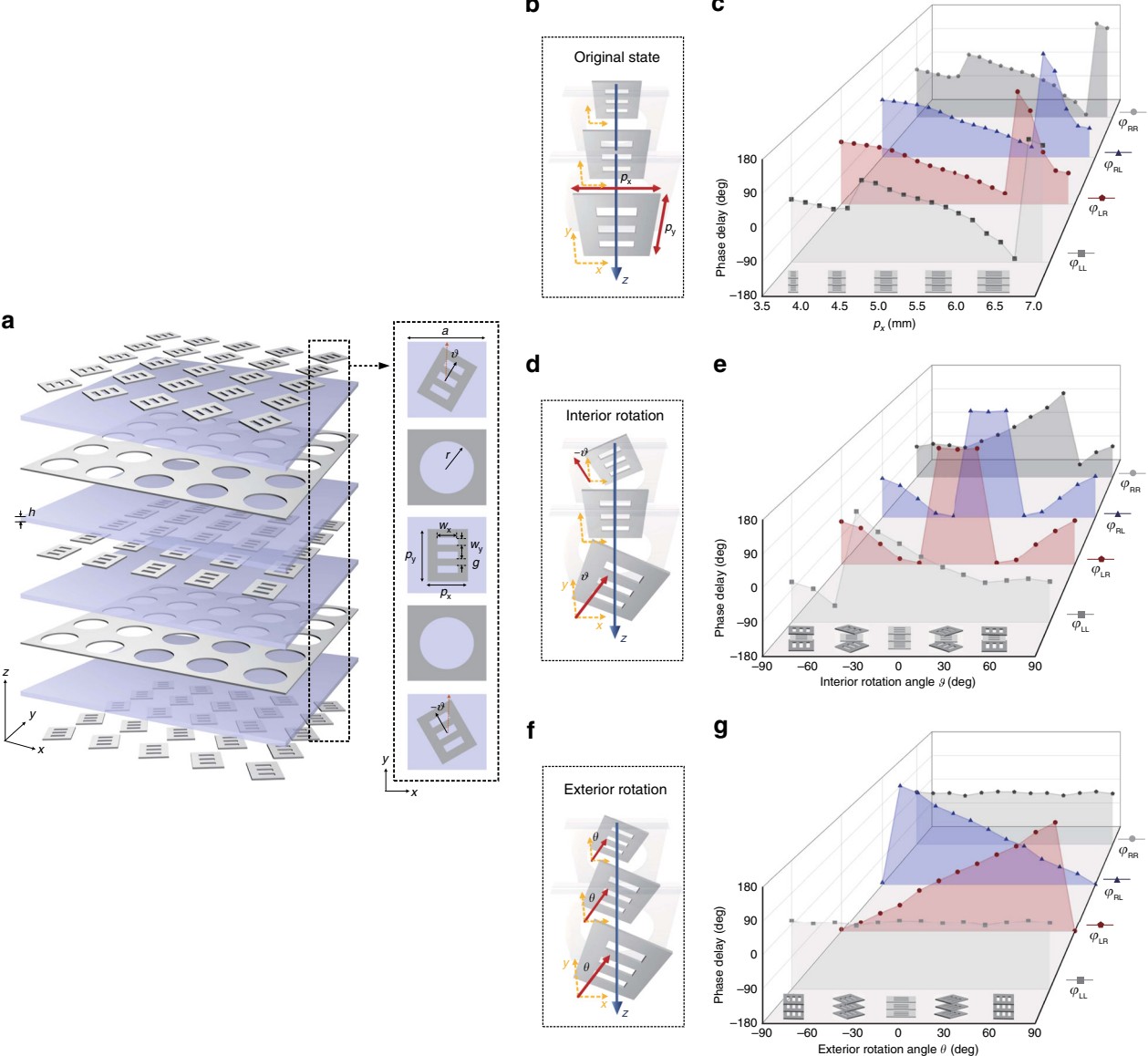

**Fig. 2 Meta-atom design and verification. a** Schematic of proposed meta-atom geometric structure, where periodicity is $a = 8.8$ mm, radius of circular aperture in the grid layer is $r = 3$ mm, thickness of each dielectric substrate is $h = 1$ mm, $p_x$ and $p_y$ are the length and width of metallic patch, $w_x$ and $w_y$ are the dimensions of rectangular gap in the metallic patch, $g$ is the distance between two adjacent gaps, and $\vartheta$ is the interior rotation angle between three patch layers. **b** 3D view of meta-atom with original state and **c** the output phase profiles of four channels against length $p_x$ of the patch layer for verifying propagation phase. **d** 3D view of meta-atom with only interior rotation and **e** the output phase profiles of four channels against interior rotation angles for demonstrating chirality-assisted phase. **f** 3D view of meta-atom with only exterior rotation and **g** the output phase profiles of four channels against exterior rotation angles for demonstrating geometric phase.

distinct tendency $\varphi_{LL} \neq \varphi_{RR}$, while the two cross-polarized phase responses keep same tendency with the interior rotation $\varphi_{LR} = \varphi_{RL}$, which is due to the symmetrical rotation angle in upper and lower patch layers (derived and explained in Supplementary Note 2 and Supplementary Note 3). It is verified that the chirality-assisted phase can decouple the intrinsic coherence between co-polarized output under orthogonal CP states, while makes no disparate effect in two cross-polarized channels. As for the geometric phase, it is performed through rotation of the whole meta-atom structure with angle $\theta$ named exterior rotation as shown in Fig. 2f. Figure 2g presents the four output phase profiles of the representative meta-atom with specific dimensions and interior angle but various exterior angles, where two cross-polarized phases exhibit different tendency $\varphi_{LR} \neq \varphi_{RL}$ and two co-polarized phases are the same $\varphi_{LL} = \varphi_{RR}$.

Based on all above theoretical and simulated results, it can be confirmed that all four phase responses of the CP channels can be separately tuned with independent geometrical parameters of the proposed meta-atom. For meta-devices pursuing complete manipulation of the four CP conversion channels to achieve multiple functionalities simultaneously, the meta-atom library to endow the required phase gradients of all CP conversion channels, is established with sweeping of length $p_x$, width $p_y$ and interior angle $\vartheta$ presented in Supplementary Note 4. Once the required phase profiles in different channels are confirmed, meta-atoms can be selected from the library and further optimized to construct the desired multifunctional metasurfaces. It is noted that, in this scheme we applied four parameter degrees of freedom ($p_x$, $p_y$, $\vartheta$, and $\theta$) to independently manipulate four phase patterns of transmission channels ($\varphi_{LL}$, $\varphi_{LR}$, $\varphi_{RL}$, and $\varphi_{RR}$). However,

within this phase-modulation process, the amplitude responses are not considered and the efficiency is partly sacrificed due to the particularity of the proposed meta-atom structure. This limitation is detailed in Supplementary Note 5 and Supplementary Fig. 4. The exploration of an extra degree of freedom (e.g., material losses, active components, and even time) for independent amplitude modulation is therefore desired in future works.

**Simulation and measurement results on two metasurfaces.** Based on the above specific explanation of general principle and construction of the meta-atom library, here we briefly clarify the construction of a single metasurface to achieve a distinct wavefront from each of the four CP channels. It is supposed that the desired spatial phase distributions can be described by $F_{LL}(x, y)$, $F_{LR}(x, y)$, $F_{RL}(x, y)$, and $F_{RR}(x, y)$ respectively. To implement the synthesized three phase schemes, chirality-assisted phase, propagation phase and PB phase components should be established by desired spatial phase distributions simultaneously (detailed in Supplementary Note 3). Following the phase modulation scheme, here we construct two metasurfaces with distinct functionalities to demonstrate the proposed phase responses for independent manipulation of full CP conversion channels.

As a first device concept, a meta-deflector able to separate all the co-polarized and cross-polarized output beams to four independent directions, is designed and the functional schematic is shown in Fig. 3a. The phase distribution of the proposed meta-deflector is based on the generalized Snell's Law and is described as:

$$\Phi_i^{\text{refr}}(x, y) = \Delta\Phi_i \cdot x = \frac{2\pi}{\lambda_0}(\sin\varsigma_i^{\text{out}} - \sin\varsigma^{\text{in}}) \cdot x,$$
$$\text{with } i = 1, 2, 3, \text{ and } 4, \qquad (3)$$

where the normal direction is set along $+z$-axis and $\Delta\Phi_i$ represents the phase gradient along $x$-axis. $\varsigma^{\text{out}}$ and $\varsigma^{\text{in}}$ are the angles of output and incident waves relative to $+z$-axis, and we consider the normal incident case where $\varsigma^{\text{in}} = 0°$. In this section, the output tilting angles are preset as $\varsigma_1^{\text{out}} = -35°$ in L–L channel, $\varsigma_2^{\text{out}} = 0°$ in L–R channel, $\varsigma_3^{\text{out}} = 58°$ in R–L channel and $\varsigma_4^{\text{out}} = -16°$ in R–R channel. The corresponding required phase gradients are $\Delta\Phi_1 = \frac{\pi}{3}$ for $F_{LL}(x,y)$, $\Delta\Phi_2 = 0$ for $F_{LR}(x,y)$, $\Delta\Phi_3 = -\frac{\pi}{2}$ for $F_{RL}(x,y)$, and $\Delta\Phi_4 = \frac{\pi}{6}$ for $F_{RR}(x,y)$. As for the construction of metasurface, the desired phase gradients for different channels along $x$-axis are discretized and implemented by 25 meta-atoms. The theoretical phase profiles and simulated phase value of all these meta-atoms are exhibited in Fig. 3b, indicating the feasibility of metasurface construction. Figure 3c displays the simulation and measurement results of the meta-deflector for four independent refracted wavefronts, whose far-field patterns at the center frequency of 10 GHz and the results clearly describe the peak direction of the output field intensities. It can be seen that under LHCP wave illumination, the co-polarized output wavefront tilts to $-35°$ (in L–L channel), while the cross-polarized component indeed does not show any refraction effect (in L–R channel). Upon flipping the incidence into RHCP, the LHCP and RHCP output components are reconstructed with deflected wavefronts of $58°$ and $-15°$ as verified by simulation and measurement (in R–L and R–R channel), respectively. It is worth-noting that in this part, the cross-polarized output wavefront under LHCP incident wave is preset with refraction angle of $0°$, indicating that the energy in this channel is manipulated with converted polarization and kept unchanged wavefront state with the input. Another meta-deflector design is presented in the Supplementary Note 8 and Supplementary Fig. 6 to verify that all four output wavefronts can be imposed with different and arbitrary refraction angles. For further

demonstration, the meta-deflector is fabricated and experimentally measured in microwave region as detailed in "Methods" section, A photograph of the sample is shown in Fig. 3d and measured results are displayed in Fig. 3e. The far-field pattern measurements have further been performed in frequency band from 9 to 11 GHz, which are function against frequency and detection angles. It can be seen that the deflecting angle slightly changes within the whole bandwidth, which can be considered as the frequency dispersion in Eq. (3). However, at the center frequency of 10 GHz, all the simulated and measured wavefronts generated in channels L–L, L–R, R–L, and R–R present the desired refraction directions, verifying the availability of proposed mechanism for simultaneous and independent manipulation of full spin conversion channels.

Additionally, we further examine the evolution of diffraction order against the variation of incident polarization state. Any arbitrary polarized wave $\kappa$ can be considered as the superposition of two orthogonal CP waves with different proportionality coefficient, as described as:

$$\kappa = \eta_L \cdot |L\rangle + \eta_R \cdot |R\rangle, \qquad (4)$$

where $\eta_L$ and $\eta_R$ denote the coefficient of LHCP and RHCP components. Here, we select five different polarization states as the incident wave, and the polarization variation path can be described as the black line on the Poincaré sphere in Fig. 4. From the north pole (point A) representing RHCP to the south pole (point E) for LHCP, continuous evolution is experienced through right-handed elliptical polarization (RHEP, point B), linear polarization ($x$-LP, point C), and left-handed elliptical polarization (LHEP, point D), where the ratios between $\eta_L$ and $\eta_R$ of these five polarized incidence A, B, C, D, and E are 0, 0.3, 1, 3, and $\infty$.

The simulated and measured output far-field intensities for five input polarization states are exhibited in Fig. 4. When the input polarization is RHCP, the output far-field intensity contains two main lobes, co-polarized component R–R with $-16°$ refraction and cross-polarized component R–L with $58°$ refraction, as shown in Fig. 4a. With the polarization of incidence changing to state B, it can be seen from Fig. 4b that the output wavefront includes four lobes located at approximately, $-35°$, $0°$, $58°$, and $-16°$ directions, which are the individual contributions of the four output channels L–L, L–R, R–L, and R–R. The proportion between the sum of energy distributed in both L–L and L–R channels to that in R–L and R–R channels is approximately 1:3, corresponding to the ratio between LHCP and RHCP component at the input end with RHEP state. Figure 4c displays the far-field intensity under LP wave illumination, where the far-field lobes in L–L and L–R channels exhibit nearly equal amplitude to that in R–L and R–R channels. When the input state is switched to LHEP, the proportion of output sum energy in both L–L and L–R channels to that in R–L, R–R channels is changed to 3:1 and displayed in Fig. 4d, as reversely symmetric situation to that in RHEP results. Finally, under LHCP wave illumination, only two main lobes with $-35°$ and $0°$ refraction are left as in Fig. 4e. According to the above evolution process, it can be confirmed that all four CP channels can be fully utilized, and the ratio between different diffraction orders can be considered as function of the incident polarization.

For further demonstration of complete manipulation of the four CP conversion channels, a spin-to-orbital angular momentum meta-convertor is proposed and designed, where the orbital angular momentum (OAM) with different topological charge is achieved in each CP channel. The spatial helical phase distribution of optical spiral phase plate for OAM generation is described as:

$$\Phi_i^{\text{OAM}}(x, y) = l_i \cdot \arctan(y/x) \text{ with } i = 1, 2, 3, \text{ and } 4, \qquad (5)$$

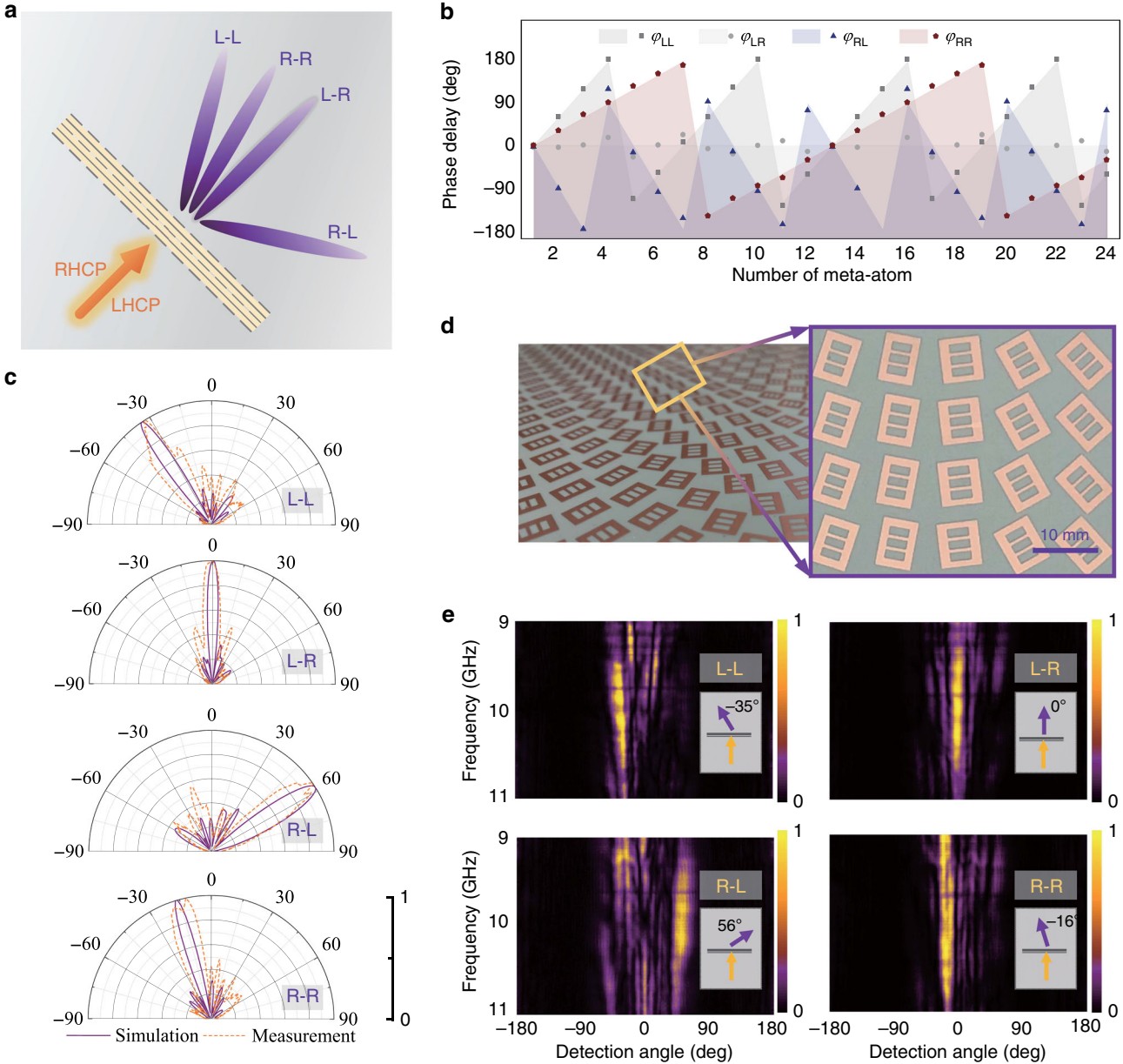

**Fig. 3 Numerical and experimental results validating the performances of the meta-deflector. a** Schematic of proposed meta-deflector. **b** Theoretical and simulated phase profiles of meta-atoms located along x-axis to provide required phase gradients in all four CP transmission channels. **c** Simulated and measured normalized far-field patterns for channels L–L, L–R, R–L, and R–R at 10 GHz, respectively. **d** Photograph of fabricated sample of proposed meta-deflector. **e** Measured normalized far-field intensities against frequency bandwidth (9–11 GHz) and detection angles for channels L–L, L–R, R–L, and R–R, respectively.

where $(x, y)$ is the required coordinate of meta-atom and $l_i = i - 1$ is the topological charge of OAM mode. $\Phi_i^{OAM}(x, y)$ with $i = 1$, 2, 3, and 4 are the corresponding phase distributions for $F_{LL}(x, y)$, $F_{LR}(x, y)$, $F_{RL}(x, y)$, and $F_{RR}(x, y)$, respectively.

Figure 5a displays the simulated energy distributions in *xoy* plane at $z = 150$ mm (corresponding to $5\lambda_0$ for the center frequency of 10 GHz) of transmitted output vortex beams carrying OAM modes with $l = 0$, 1, 2, and 3 through the CP channels L–L, L–R, R–L, and R–R orderly, where all the typical doughnut-shape energy rings are exhibited and compared. It can be obviously seen that in L–L channel, the OAM mode is 0, whose energy intensity is as a plane wave with no hollow distribution in the center. Meanwhile with the topological charge increasing from $l = 1$–3 in L–R, R–L, and R–R channels, respectively, the radius of energy rings is gradually enlarged, which experimentally

verifies the divergent properties of typical vortex beam. Figure 5b presents the corresponding measured energy distributions of vortex beams generated from the CP conversion channels, exhibiting four vortex beams with increasing energy ring radius, which is in good agreement with the simulations. Meanwhile, it can be seen that there exist some nonuniform and discrepancies in simulated and measured energy distributions, which are detailed discussed in Supplementary Note 6. The simulated and measured phase distributions of vortex beams carrying the corresponding four OAM modes are shown in Fig. 5c, d. It can be obviously observed that the helical phase pattern of OAM modes are totally distinct, verifying that the functionalities in all four CP channels can be independently modulated and the helical patterns are in accordance with the topological charge $l \cdot 2\pi$. The measured phase patterns agree well with the simulated results,

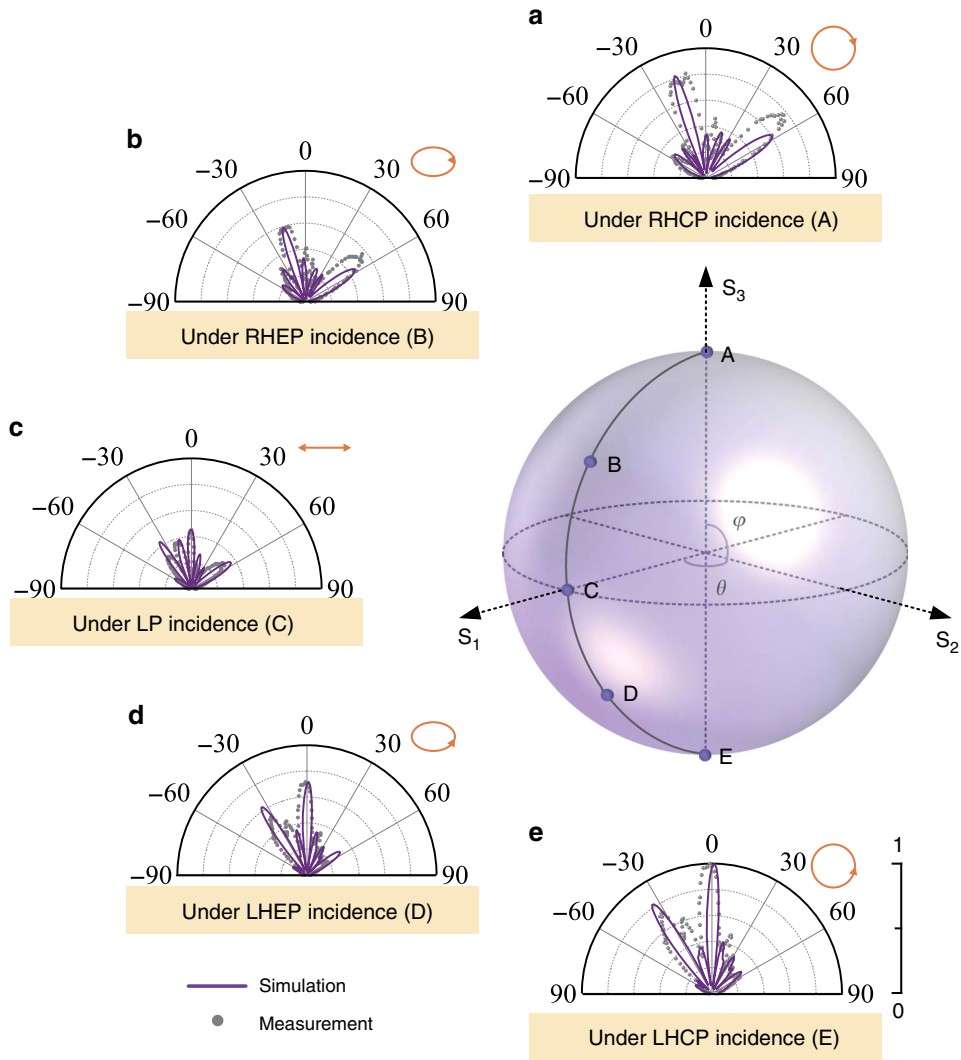

**Fig. 4 Evolution of diffraction order against the variation of incident polarization state on the Poincaré sphere. a** Under the RHCP incidence, two main output lobes located at −16° and 58° carry co-polarization and cross-polarization respectively. **b** Under the RHEP incidence, four lobes appears at −35°, −16°, 0°, and 58° for L–L, R–R, L–R, and R–L channels respectively. **c** Under the LP incidence, four lobes are exhibited at −35°, −16°, 0°, and 58°. **d** Under the LHEP incidence, four lobes are shown at −35°, −16°, 0°, and 58°, which is opposite to that of RHEP incidence. **e** Under the LHCP incidence, only two main output lobes located at −35° and 0° carry co-polarization and cross-polarization, respectively.

which further indicates the feasibility of independent multiple OAM modes generation based on full CP conversion channels.

Here, it should be noted that the phase manipulation to achieve distinct responses in all four transmission CP channels is a general method, which is not limited by operating wavelength. To extend the method to shorter wavelengths, efforts in high precision lithography would be required. Moreover, metallic lossy plasmonic resonators to be used in the replacement of current metallic patches, would have to be replaced by chiral all-dielectric nanoantennas.

## Discussion

We propose a general criterion to construct metasurfaces, which activate all CP channels and make full utilization of transmitted energy simultaneously. Through the synthesis of chirality-assisted phase, propagation phase and geometric phase, all the components in the Jones matrix can be decoupled and independently tuned. Two meta-devices are proposed based on synthesized three phase modulations to achieve the multifunctional wavefronts through four CP channels. Additionally, the evolution of diffraction orders of output far-field patterns against the variation of

input polarization state is verified, further indicating the feasibility of proposed schemes for full channels utilization. This paradigm can be further extended and applied to other frequency spectrum with appropriate chiral meta-structures.

## Methods

**Microwave sample fabrication**. The prototypes proposed in this paper are fabricated by using classical printed circuit board (PCB) technique, and which is briefly introduced as follows. In fabrication process, meta-structures are integrated with four pieces of polyfluortetraethylene dielectric-slabs and three adhesive layers. The 1 mm-thick dielectric substrates used have a relative permittivity $\varepsilon_r = 3.5$ and double copper-cladding layer of 0.035 mm-thick, while the adhesive layer has a relative permittivity $\varepsilon_r = 2.74$ and thickness of 0.1 mm. The total thickness of fabricated samples is 4.475 mm (about 0.15 $\lambda_0$ at 10 GHz). The proposed metasurfaces all consist of 25 × 25 meta-atoms, which exhibit total size of 220 mm × 220 mm, and the fabrication tolerance is in accordance with initial design requirements.

**Experimental measurements**. Measurements are conducted by a setup surrounded by microwave absorbers in order to minimize parasitic reflections. A 2–18 GHz dual-polarized wideband horn antenna is used as the feeding source to launch the circularly polarized quasi-plane waves (left-handed circular polarization, and right-handed circular polarization). For the near-filed mapping measurement, a fiber optic active antenna is used as field probe to measure both the amplitude and

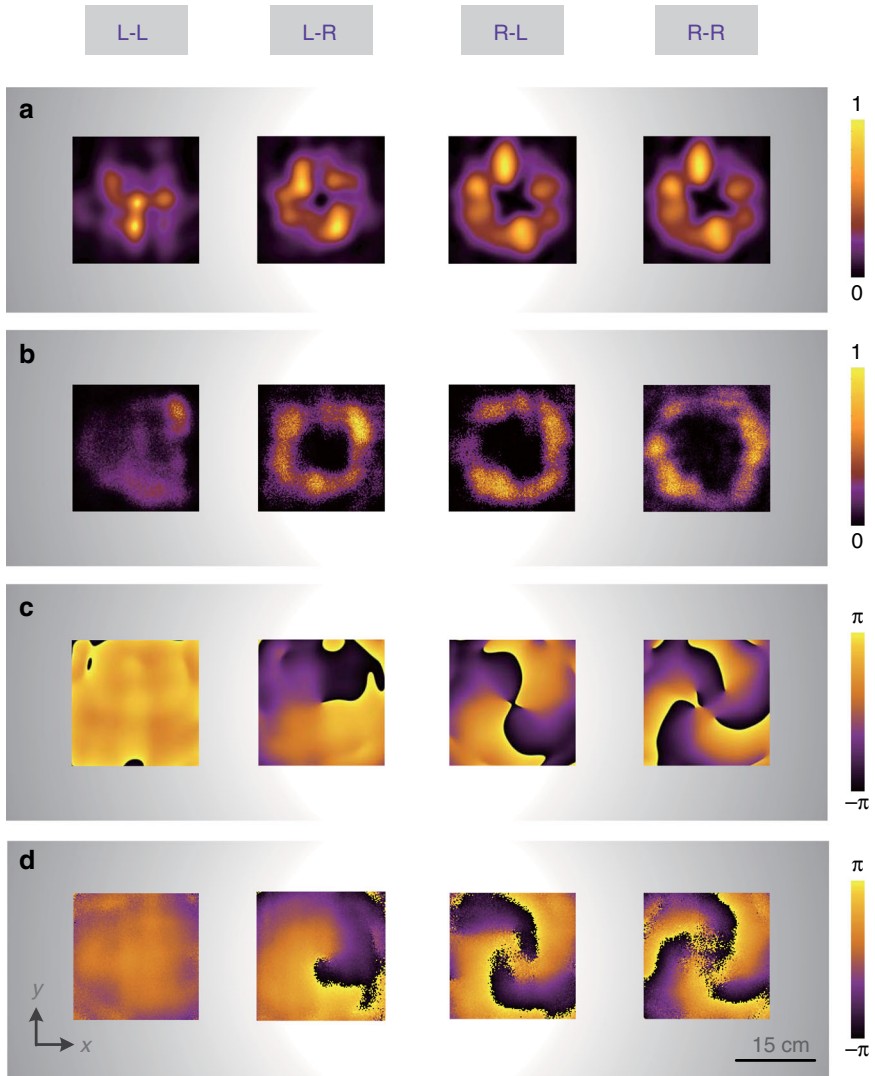

**Fig. 5 Demonstration for proposed spin-to-orbital angular momentum meta-convertor. a** Simulated and **b** measured energy intensities, and **c** simulated and **d** measured phase distributions of output vortex beams carrying different OAM modes in *xoy* plane with $z = 150$ mm.

phase of the electric field. The probe receiver is fixed on two translation stages controlled by a motion controller and its position is incremented by a step of 2 mm. For far-field pattern measurement, the receiver is set as a horn antenna that is similar to the transmitting antenna. The fabricated sample and transmitting antenna are placed on a turntable to provide an angular scanning, the receiving antenna is fixed and placed at a distance $d > 20\lambda_0$ away from the meta-device. In both measurement systems, the transmitting and receiving antennas are connected to an Agilent 8722ES vector network analyzer, which is adopted to measure the complex $S_{11}$ and $S_{21}$ parameters including the amplitude and phase. The schematic of microwave experimental setups and measurement calculation methods are shown in Supplementary Note 7 and Supplementary Fig. 5.

## Data availability

The data that support the plots within this paper and other findings of this study are available from the corresponding author upon reasonable request.

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

## Acknowledgements

K.Z. acknowledges funding from National Natural Science Foundation of China under Grant No. 61771172. Q.S. and P.G. acknowledge funding from the European Research Council (ERC) under the European Union's Horizon 2020 research and innovation programme (Grant agreements no. 639109).

## Author contributions

K.Z., S.N.B., and P.G. conceived the idea. Y.Y. and B.R. conducted the numerical simulations and theoretical analysis. K.Z., Y.Y., Q.S., S.N.B., X.D., Q.W., and P.G. wrote the manuscript. All authors participated in the experiments and data analysis and read the manuscript.

## Competing interests

The authors declare no competing interests.
