## [Peer Review File · Nature Communications]

REVIEWER COMMENTS

Reviewer #1 (Remarks to the Author):

This paper introduces a chirality-based degree of freedom to enable more flexible control of the phase changes when a microwave propagates through a metasurface. A metasurface has four output channels, namely RHCP output under LHCP or RHCP input and the LHCP output under the LHCP or RHCP input. The major claim of this paper is that by introduce the chirality-based phase change, one can control the phase changes of the four channels independently.

While the motivation is sound, the idea of using multiple layers to introduce more degree of freedom (and its mathematical derivation) is straightforward. I therefore have reservation about its novelty.

I also have a concern. It seems that the four degrees of freedom to control the phase changes of the four output channels rely on the control $T_{\{LL\}}$, $T_{\{LR\}}$, $T_{\{RL\}}$, and $T_{\{RR\}}$. The derivation of Eq. (S5c) shows that the phase change introduced by $T_{\{LR\}}$ and $T_{\{RL\}}$ are the same. If they are the same, then according to Eq. (2), $t_{\{xy\}}-t_{\{xy\}}$ may vanish, and you loose one degree of freedom. I understand that due to the cross-layer coupling, $T_{\{LR\}}$ and $T_{\{RL\}}$ are not the same. But to me, this is a limitation: First, I'm not sure to what extent $T_{\{LR\}}$ and $T_{\{RL\}}$ are different due to coupling. Secondly, it is unclear how to precisely control the difference between $T_{\{LR\}}$ and $T_{\{RL\}}$ by changing the system parameters.

I have another concern about the fabrication of the multi-layer metasurface. This work demonstrates it in microwave so the antenna is relatively large. But when applying it to make optical metasurfaces, manufacturing these layers and the alignment across multiple layers of optical metasurfaces can be a serious hurdle, as far as I know. This would largely limit the applicability of the proposed method.

Minor comments:

I think the authors need to proof read their draft more carefully. There are quite a few typos or grammar issues. here are a few examples (among others):

- Line 81: meatsurfaces -> metasurfaces
- Line 114: to decouple inherent consistent -> to decouple inherent consistency?
- Line 58 of supplementary: the subscripts of α should be 1, 2, 3.
- In eq. (S3), I believe the bra and ket in each line should be switched.
- Line 64 of supplementary: "amplitude of co-polarized and cross-polarized are" not grammatically correct.
- Line 77 of supplementary: "independent perfect wave plates" you mean "perfectly independent wave plates"?

Reviewer #2 (Remarks to the Author):

While past work has demonstrated arbitrary symmetry and unitary Jones matrices operating in transmission mode, the authors demonstrate extended functionality by including chirality to construct Jones matrices that are not symmetric. They show that by controlling the propagation phase, geometric phase, and chirality, the phase of four distinct channels of circularly polarization transmitting through the device may be controlled. The results and approach are non-obvious and impressive, and of great interest to the community. The presentation of the results is excellent, though the English could use some work. The explanation lacks clarity in places, has a few inaccuracies, and is burdened by poor notation, meaning the pedagogical worth of the paper is lacking. Assuming these issues and the following two major corrections are addressed, I am happy to recommend this work for publication in Nature Communications. I detail major and minor issues

below.

Major issues whose resolution I consider necessary for publication:

1. The assumptions on the form of the (4x4) scattering matrices of each meta-atom being achieved here should be stated explicitly. It appears from their description that the authors assume a reciprocal, lossless, matched system. In this case, the scattering matrix is symmetric and unitary, and there is negligible reflection (impedance matched). If this is true, two statements the authors make are demonstrably false. If this is not true, the authors should clarify and address how this resolves the following problems:

(a) The authors claim that the four transmission coefficients in the Jones matrix are totally independent. This is not true if the system is reciprocal (passive elements only), lossless (no material losses), and matched (i.e. no reflection losses either). It is easy to show that $\text{abs}(t_{xx}) = \text{abs}(t_{yy})$ and $\text{abs}(t_{xy}) = \text{abs}(t_{yx})$ in this case. The phases of each component, however, are indeed independent in principle (the complex Jones matrix need not be symmetric). In transmission mode, reciprocity constrains the phase of t_{xy} from side 1 to 2 to be the same as t_{yx} from side 2 to 1, but does not constrain the phases of t_{xy} and t_{yx} when incident from the same side. This inaccuracy does not change the authors' results, since they focus on phase manipulation and choose all of these amplitudes to be equal to each other anyways. Therefore, this is easily fixed by clarifying the relevant passages to specify that the phases are independent.

(b) The authors claim that the feat can be achieved in Reflection mode as well (where the matched assumption is omitted, of course, and presumably a mirror eliminates the transmission-side ports). This is not true if the system is reciprocal, because $r_{xy} = r_{yx}$ in this case. In other words, the complex Jones matrix is symmetric in reflection mode by reciprocity. That is, both the amplitude and phase of the off-diagonal components of the reflected Jones matrix are identical. There simply aren't four independent phases to control in a reciprocal mirror with one diffraction order. Again, this does not change the authors' results for transmission mode. However, it is a basic inaccuracy that should be remedied before publication.

2. The relationship between dependent and independent degrees of freedom is not made completely clear, even after reading the main text and supplementary several times. For a reciprocal, lossless, matched system, there are 5 degrees of freedom in the transmitted Jones matrix: four phases, and one relative amplitude (ratio between a diagonal component and an off-diagonal component). The authors vary four geometric degrees of freedom: p_x , p_y , internal angle, and external angle. By using one of these degrees of freedom, one may easily set the amplitude of diagonal and off-diagonal components to be equal. It is implied that the degree of birefringence is fixed to set the amplitude to be equal, and the uninquisitive reader is thereby led to believe that this is achievable for any choice of the four phases. But this constraint uses up a degree of freedom (e.g. it fixes p_y for a choice of p_x and internal angle). This means there are three degrees of freedom remaining to control the desired four phases (in each the LL,RR,LR,RL channels), which is not feasible even in principle.

Alternatively, the constraint on equal amplitude may be relaxed so that the four phases may be controlled independently. In supplementary section S5, the authors briefly discuss how the transmission amplitude of each component is not, in fact, perfectly equal. The authors explain this by the fact that the underlying phase-control mechanism is a resonance, which does not have constant amplitude as a function of p_x and p_y . However, this complication notwithstanding, it is not discussed in the manuscript whether the amplitudes can be equal even in principle (e.g. by using dielectric structures which impart phase in a non-resonant manner). The simple degrees-of-freedom argument above suggests that it is not possible without a fifth degree of freedom.

This limitation should be clarified explicitly, and the impact on achievable devices should be discussed. The fact that the amplitude of each channel is not independent of the chosen phases is highly relevant to experts in the field. Great effort was made in the initial metasurface works (e.g.,

the classic Ref [8]) to maintain constant amplitude, so that the phase response alone controls the local diffraction efficiencies across the device. This is a key assumption behind the Generalized Snell's Law (Ref [8]), used by the authors in Equation (3), wherein a single diffraction order is selected for locally. A direct result of this amplitude limitation in the authors' case is that the diffraction efficiencies into the desired channels may not all simultaneously be 50% (the ideal case for the authors) for arbitrary phase functions even in principle. The impact on diffraction efficiencies due to amplitude variations may partially be compensated, as briefly mentioned by the authors in the supplementary section S5, but not perfectly.

I personally do not think this limitation of the design is enough to warrant rejection from Nature Communications, though it means the claims of the paper need to be clarified. In my view, the results speak for themselves: reasonable control over four output channels is achievable, which is in my view a significant advance to the field. It is well-known that phase is more important than amplitude in diffractive devices, so the variance of amplitude as a cost of control of 4 phase is a worthwhile, if somewhat limited, achievement. However, the implied assumption behind the vast majority of metasurface works is that the efficiencies could be ideal in principle, and are limited in practice by various imperfections in the control of optical responses and fabrication errors. Future work solving these practical and engineering imperfections are then immediately inheritable. Here, on the other hand, the efficiencies are limited by construction: it is a limitation in the design approach itself. This difference is a crucial to spell out, and a paper that hides it rather than clarifies it should be rejected. I think it is appropriate to say that future work can add that 5th degree of freedom to fully control the light, while the current manuscript represents a step to that final goal.

Minor issues that I suggest resolving to improve the manuscript:

1. In Figure 5, the use of a black background to organize the data is inappropriate if black is included in the colormap representing the data. It is sometimes difficult to tell where the data ends and where the background starts. A drawn border indicating these edges would also fix this issue without changing the background black color.
2. The notation (e.g. $E_{(LHCP,inc)}^{(out-co)}$) is needlessly cumbersome, and hinders comprehension. Why not use $E_{LL}, E_{RL}, E_{LR}, E_{RR}$ instead? This immediately tells the reader all the relevant information. It is readily apparent that E_{RL}, E_{LR} are cross-polarized components while E_{LL}, E_{RR} are co-polarized components.
3. The matrices $T1$ and $T2$, defined in Equation (1), are never again referred to.
4. Related to point 2 in the "major issues" list above, the co and cross phase terms in Equation S5 appear as separate, while the discussion around Equation S2 specifically mentions that they are the same. Why do they appear separate in Equation S5?
5. Relatedly, a closed-form relating the phases of $E_{LL}, E_{RL}, E_{LR}, E_{RR}$ to the phases of t_{xx} and t_{yy} , and the internal and external angle, would be highly valuable. Derivation of such a relationship would also show the amplitude variation discussed in point 2 above.
6. Bold typeface appears to be used for both matrices and the elements of matrices, which is non-standard.
7. In Fig 5b the data is misaligned in the vertical direction.
8. The title's phrase of "full wavefront modulation" is non-specific and not accurate given the 5th remaining degree of freedom being uncontrolled.

Adam Overvig

Reviewer #1 (Remarks to the Author):

Comments: This paper introduces a chirality-based degree of freedom to enable more flexible control of the phase changes when a microwave propagates through a metasurface. A metasurface has four output channels, namely RHCP output under LHCP or RHCP input and the LHCP output under the LHCP or RHCP input. The major claim of this paper is that by introduce the chirality-based phase change, one can control the phase changes of the four channels independently.

While the motivation is sound, the idea of using multiple layers to introduce more degree of freedom (and its mathematical derivation) is straightforward. I therefore have reservation about its novelty.

Response: The authors are grateful for the comments. Here we wish to explain the novelty of this work. Firstly, as for the previous chiral-metasurface works, most of the works are devoted to the amplitude response modulation, namely circular dichroism (CD). In this paper, the phase responses introduced by structure chirality is analyzed and modulated, which is then further utilized as an additional degree of freedom to impose required wavefronts. By synthesis of chirality-based phase, propagation phase and geometric phase, all four output channels can be fully manipulated simultaneously and independently. To the best of authors' knowledge, it is the first time that the combination of these three degrees of freedom for phase modulation (chirality-assisted phase, propagation phase and geometric phase) is proposed. Here, the three phase resources acting as degrees of freedom produce distinct influences in the four output channels, and all the decoupling process of intrinsic consistencies among the four output channels are analyzed in detail and successfully demonstrated. Two metasurfaces are proposed to create different wavefronts in four transmission channels respectively, verifying the feasibility of proposed mechanism. The novelty of this work is specially emphasized and added in red color in the revised introduction of main text.

Additionally, this mechanism can generally be extended to other operating wavelength regimes as long as appropriate meta-atom structure can be selected to simultaneously implement the three phase responses by tailoring corresponding parameters. Multi-layered structure used in this paper is just a demonstration of the proposed full phase-modulation scheme. By the utilization of all four transmission channels, the functionalities of one single meta-device can be boosted to be versatile. The proposed scheme for independent phase-manipulation is an alternative for wavefronts multiplexing and multi-functionality applications.

Added or Revised Text:

In this work, a general phase modulation scheme for the construction of metasurfaces enabling distinct wavefronts in all four CP channels is established. Chirality-assisted phase response, instead of amplitude response (namely circular dichroism), is proposed and then synthesized with propagation phase and PB phase to activate and distinguish wavefront engineering in four co- and cross-polarized channels under LHCP and RHCP wave illuminations.

Comments: I also have a concern. It seems that the four degrees of freedom to control the phase changes of the four output channels rely on the control $T_{\{LL\}}$, $T_{\{LR\}}$, $T_{\{RL\}}$, and $T_{\{RR\}}$. The derivation of Eq. (S5c) shows that the phase change introduced by $T_{\{LR\}}$ and $T_{\{RL\}}$ are the same. If they are the same, then according to Eq. (2), $t_{\{xy\}} - t_{\{yx\}}$ may vanish, and you lose one degree of freedom. I understand that due to the cross-layer coupling, $T_{\{LR\}}$ and $T_{\{RL\}}$ are not the same. But to me, this is a limitation: First, I'm not sure to what extent $T_{\{LR\}}$ and $T_{\{RL\}}$ are different due to coupling. Secondly, it is unclear how to precisely control the difference between $T_{\{LR\}}$ and $T_{\{RL\}}$ by changing the system parameters.

Response: As mentioned in this comment, there are four parameter degrees of freedom: p_x , p_y (width and length of the rectangular patch) for propagation phase modulation, ϑ (interior rotation angle) for chirality-assisted phase modulation, and θ (exterior rotation angle) for geometric phase modulation. Generally, four elements of transmission matrix own some inherent consistency with each other, and we apply these physical degrees of freedom to decouple the responses and achieve independent wavefronts in four channels. To explain the modulation mechanisms, let's consider meta-atom designed with fixed p_x and p_y , $\vartheta = 0$ and $\theta = 0$. Its responses of T_{LL} and T_{RR} are identical, while T_{LR} and T_{RL} are also equal. From this situation, the interior rotation angle ϑ can be introduced to decouple the consistency between T_{LL} and T_{RR} while keeping T_{LR} and T_{RL} unchanged. Similarly, the exterior rotation angle θ can further realize distinct responses of T_{LR} and T_{RL} , meanwhile producing no effect on T_{LL} and T_{RR} .

Eq. (2) in the main text presents the output electric field based on the general Jones matrix of a transmission-type meta-structure, which has already taken exterior rotation θ into consideration. Eq. (S5) is the four output phase expressions derived from the meta-atom structure with specific interior rotation and no exterior rotation ($\vartheta \neq 0$ and $\theta = 0$), which is established to show the effect on phase responses produced by the chirality of the meta-atom structure. The reviewer is right that phase change

introduced by $T_{\{LR\}}$ and $T_{\{RL\}}$ are the same in Eq. (S5c). The phase relationship in Eq. (S5c) can be mapped to Eq. (2) with $\theta = 0$ and we can obtain:

$$\begin{aligned}\phi'_{LR} &= \arg[(T_{xx} - T_{yy}) - i \cdot (T_{xy} + T_{yx})] \\ \phi'_{RL} &= \arg[(T_{xx} - T_{yy}) + i \cdot (T_{xy} + T_{yx})] \\ \phi'_{LR} &= \phi'_{RL}\end{aligned}\quad (R1)$$

From Eq. (R1) we can know that, if $T_{xy} + T_{yx} = 0$ and $T_{xy} - T_{yx} \neq 0$, the condition $\phi'_{LR} = \phi'_{RL}$ can be satisfied. However, when the exterior rotation ($\theta \neq 0$) is considered as we do in Eq. (2), the total output phase of T_{LR} and T_{RL} channels are described as:

$$\begin{aligned}\phi_{LR} &= \phi'_{LR} + 2\theta \\ \phi_{RL} &= \phi'_{RL} - 2\theta\end{aligned}\quad (R2)$$

It can be seen from Eq. (R2) that, the degree of freedom for manipulating T_{LR} and T_{RL} is the exterior rotation angle θ , which would introduce opposite phase interruptions in T_{LR} and T_{RL} channels respectively. Due to the rotation characteristics, any phase difference (from 0 to 2π) can be imposed into these two channels. For all these manipulations, there is no degree of freedom lost.

In order to illustrate the working principle of proposed meta-atom clearer, we have re-organized the equations and re-clarified the principles in revised main text and Supporting Information in red color, and correspondingly changed the complicated notations.

Added or Revised Text:

i). In main text

Different from previous works on birefringent metasurfaces, the aim of this work is to perform distinct phase-modulation in the four CP transmission channels, which can be described by the four element

transmission coefficients in the Jones matrix $T_{cir} = \begin{bmatrix} t_{LL} & t_{LR} \\ t_{RL} & t_{RR} \end{bmatrix}$. The equivalent metasurface system is

supposed to be passive, lossless, matched and reciprocal. Therefore, the four CP transmission coefficients (the first / second subscript represents the input / output CP state, L / R denotes the LHCP / RHCP state) with linear base can be described as follows:

$$t_{LL} = \frac{1}{2}[(t_{xx} + t_{yy}) + i \cdot (t_{xy} - t_{yx})] \quad (1a)$$

$$t_{LR} = \frac{1}{2}[(t_{xx} - t_{yy}) - i \cdot (t_{xy} + t_{yx})] \cdot e^{i2\theta} \quad (1b)$$

$$t_{RL} = \frac{1}{2}[(t_{xx} - t_{yy}) + i \cdot (t_{xy} + t_{yx})] \cdot e^{-i \cdot 2\theta} \quad (1c)$$

$$t_{RR} = \frac{1}{2}[(t_{xx} + t_{yy}) - i \cdot (t_{xy} - t_{yx})] \quad (1d)$$

where $t_{xx} = |t_{xx}| \cdot e^{i\varphi_{xx}}$ and $t_{yy} = |t_{yy}| \cdot e^{i\varphi_{yy}}$ are the diagonal linear transmission coefficients, and $t_{xy} = |t_{xy}| \cdot e^{i\varphi_{xy}}$ and $t_{yx} = |t_{yx}| \cdot e^{i\varphi_{yx}}$ are the off-diagonal linear transmission coefficients. θ is the exterior

rotation angle introduced by rotation matrix $M(\theta) = \begin{bmatrix} \cos \theta & \sin \theta \\ -\sin \theta & \cos \theta \end{bmatrix}$. Here, t_{LL} and t_{RR} are defined as

co-polarized transmission channels, which maintain the polarization state of input waves. t_{LR} and t_{RL} represent cross-polarized channels, which flip the output fields into opposite CP state. The first components with totally same expression in t_{LL} and t_{RR} (or t_{LR} and t_{RL}) can be labeled as

$t_{propa}^{co} = |t_{propa}^{co}| e^{i\varphi_{propa}^{co}} = \frac{1}{2}(t_{xx} + t_{yy})$ ($t_{propa}^{cross} = |t_{propa}^{cross}| e^{i\varphi_{propa}^{cross}} = \frac{1}{2}(t_{xx} - t_{yy})$). The phase pattern of the two

transmission components φ_{propa}^{co} and φ_{propa}^{cross} are uniquely dependent to propagation phase modulation, which would produce initial influence in both co-and cross-polarized fields regardless of the incident CP state. Meanwhile, the second components carrying opposite symbols in t_{LL} and t_{RR} (or t_{LR} and t_{RL})

can be extracted as $t_{chiral}^{co} = |t_{chiral}^{co}| e^{i\varphi_{chiral}^{co}} = \frac{1}{2} \cdot i \cdot (t_{xy} - t_{yx})$ ($t_{chiral}^{cross} = |t_{chiral}^{cross}| e^{i\varphi_{chiral}^{cross}} = \frac{1}{2} \cdot i \cdot (t_{xy} + t_{yx})$). The

phase pattern of these two components φ_{chiral}^{co} and φ_{chiral}^{cross} are determined by chirality-assisted phase, indicating that the two components would be an additional degree of freedom to decouple inherent consistency between co-polarized channels. Moreover, the PB phase pattern generated by rotating

matrix $\varphi_{geo} = 2\theta$, which is sensitive to the incidence polarization state, influence only the two cross-polarized fields. Here the plus or minus (\pm) sign represents the conjugate responses to LHCP or RHCP input states. According to Equation (1), it can be concluded that the full phase-modulation

scheme is based on three phases that constitute three degrees of freedom to decouple the inherent consistencies between CP transmission coefficients: i) propagation phase modulation is applied to define the original phase profiles of two diagonal and two off-diagonal transmission elements t_{LL} ($= t_{RR}$) and t_{LR} ($= t_{RL}$) when there is no chirality-assisted and geometric phase responses, ii) the chirality-assisted phase modulation is introduced to decouple the consistency between t_{LL} and t_{RR} while keeping t_{LR} and t_{RL} unchanged. iii) the geometric phase modulation can further realize distinct profiles of t_{LR} and t_{RL} , meanwhile producing no effect on t_{LL} and t_{RR} .

When the proposed metasurface is illuminated by LHCP incident wave $|\bar{L}\rangle = \begin{bmatrix} 1 \\ i \end{bmatrix}$ and RHCP incident wave $|\bar{R}\rangle = \begin{bmatrix} 1 \\ -i \end{bmatrix}$ respectively, the output electric fields can be expressed as:

$$\begin{aligned} \bar{E}_{out}^{L,in} &= T \cdot |L\rangle \\ &= (t_{propa}^{co} + t_{chiral}^{co}) \cdot |L\rangle + (t_{propa}^{cross} - t_{chiral}^{cross}) \cdot e^{i\phi_{geo}} \cdot |R\rangle \\ &= \bar{E}_{LL} + \bar{E}_{LR} \end{aligned} \quad (2a)$$

$$\begin{aligned} \bar{E}_{out}^{R,in} &= T \cdot |R\rangle \\ &= (t_{propa}^{co} - t_{chiral}^{co}) \cdot |R\rangle + (t_{propa}^{cross} + t_{chiral}^{cross}) \cdot e^{-i\phi_{geo}} \cdot |L\rangle \\ &= \bar{E}_{RR} + \bar{E}_{RL} \end{aligned} \quad (2b)$$

It can be observed from Equation (2) that under the orthogonal CP illuminations, the metasurface can produce four different output fields, including two co-polarized output components \bar{E}_{LL} and \bar{E}_{RR} , and two cross-polarized components \bar{E}_{LR} and \bar{E}_{RL} . Based on the analyses above, four CP transmission coefficients can be independently phase-modulated, where the inherent consistencies between four output phase patterns ϕ_{LL} , ϕ_{LR} , ϕ_{RL} , and ϕ_{RR} are independently decoupled by propagation phase, chirality-assisted phase and geometric phase, respectively.

ii). In Supporting Information

In order to simplify the cascaded Jones matrix conveniently, here, the rotation angles of each patch layer is set as $\alpha_1 = -\vartheta$, $\alpha_2 = 0$ and $\alpha_3 = \vartheta$, and the cascaded transmission expression can be obtained as:

$$\begin{aligned} T_{cas} &= \frac{1}{8}(t_{0,xx} + t_{0,yy})^3 \cdot I \\ &+ \frac{1}{8}[(t_{0,xx} + t_{0,yy}) \cdot (t_{0,xx} - t_{0,yy})^2 \cdot (2e^{i2\vartheta} + e^{i4\vartheta})] \cdot |L\rangle\langle L| \\ &+ \frac{1}{8}[(t_{0,xx} + t_{0,yy})(t_{0,xx} - t_{0,yy})^2 \cdot (2e^{-i2\vartheta} + e^{-i4\vartheta})] \cdot |R\rangle\langle R| \\ &+ \frac{1}{8}[(t_{0,xx} - t_{0,yy})^3 + (t_{0,xx} + t_{0,yy})^2 \cdot (t_{0,xx} - t_{0,yy}) \cdot (1 + e^{-i2\vartheta} + e^{i2\vartheta})] \cdot |L\rangle\langle R| \\ &+ \frac{1}{8}[(t_{0,xx} - t_{0,yy})^3 + (t_{0,xx} + t_{0,yy})^2 \cdot (t_{0,xx} - t_{0,yy}) \cdot (1 + e^{i2\vartheta} + e^{-i2\vartheta})] \cdot |R\rangle\langle L| \end{aligned} \quad (S2)$$

Based on Equation (S2), the Jones matrix of the meta-atom, which is set with the specific interior rotation and no exterior rotation, can be extracted and illustrated by T_1 :

$$\begin{aligned}
 T_1(\vartheta) &= \begin{bmatrix} T_{1LL} & T_{1LR} \\ T_{1RL} & T_{1RR} \end{bmatrix} \\
 &= \frac{1}{8} \begin{bmatrix} (t_{0xx} + t_{0yy})^3 + (t_{0xx} + t_{0yy}) \cdot (t_{0xx} - t_{0yy})^2 \cdot (2e^{i2\vartheta} + e^{i4\vartheta}) & (t_{0xx} - t_{0yy})^3 + 2 \cdot (t_{0xx} + t_{0yy})^2 \cdot (t_{0xx} - t_{0yy}) \cdot \cos 2\vartheta \\ (t_{0xx} - t_{0yy})^3 + 2 \cdot (t_{0xx} + t_{0yy})^2 \cdot (t_{0xx} - t_{0yy}) \cdot \cos 2\vartheta & (t_{0xx} + t_{0yy})^3 + (t_{0xx} + t_{0yy})(t_{0xx} - t_{0yy})^2 \cdot (2e^{-i2\vartheta} + e^{-i4\vartheta}) \end{bmatrix} \quad (S3)
 \end{aligned}$$

In order to calculate the output phase patterns in four transmission coefficients in T_1 , here it is supposed that the linear transmission amplitude for each ideal birefringent wave plate model is regarded as 1 ($|t_{0xx}| = |t_{0yy}| = 1$). The phase difference and phase combination between the two linear orthogonal directions are defined as $\Delta\varphi = \varphi_{0xx} - \varphi_{0yy}$ and $\Sigma\varphi = \varphi_{0xx} + \varphi_{0yy}$. For the further simplification and calculation of Jones matrix of equivalent meta-atom with interior rotation and no exterior rotation, the following two relationships can be obtained:

$$\frac{1}{2}(t_{0xx} + t_{0yy}) = \cos \frac{\Delta\varphi}{2} \cdot e^{i\frac{\Sigma\varphi}{2}} \quad (S4a)$$

$$\frac{1}{2}(t_{0xx} - t_{0yy}) = i \cdot \sin \frac{\Delta\varphi}{2} \cdot e^{i\frac{\Sigma\varphi}{2}} \quad (S4b)$$

By substituting Equation (S4) into Equation (S3), phase patterns of four transmission channels, introduced by the chirality responses of interior rotation angle ϑ , can be deduced as:

$$\varphi_{1LL} = \arg\left[\cos^3 \frac{\Delta\varphi}{2} \cdot e^{i\frac{3}{2}\Sigma\varphi} + \left(\cos \frac{\Delta\varphi}{2} \cdot \sin^2 \frac{\Delta\varphi}{2} \cdot e^{i\frac{3}{2}\Sigma\varphi + \pi/2}\right) \cdot (2e^{i2\vartheta} + e^{i4\vartheta})\right] \quad (S5a)$$

$$\varphi_{1RR} = \arg\left[\cos^3 \frac{\Delta\varphi}{2} \cdot e^{i\frac{3}{2}\Sigma\varphi} + \left(\cos \frac{\Delta\varphi}{2} \cdot \sin^2 \frac{\Delta\varphi}{2} \cdot e^{i\frac{3}{2}\Sigma\varphi + \pi/2}\right) \cdot (2e^{-i2\vartheta} + e^{-i4\vartheta})\right] \quad (S5b)$$

$$\varphi_{1LR} = \varphi_{1RL} = \arg\left[2 \cdot \cos^2 \frac{\Delta\varphi}{2} \cdot \sin \frac{\Delta\varphi}{2} \cdot \cos 2\vartheta \cdot \sin^3 \frac{\Delta\varphi}{2} \cdot e^{i\frac{3}{2}\Sigma\varphi + \pi/2}\right] \quad (S5c)$$

Equation (S5) expresses the chirality-assisted phase functions of four CP channels transmission coefficients. It can be seen that the phase patterns of two co-polarized channels L-L and R-R have been decoupled with different phase pattern by introducing interior rotation angle ϑ , while the cross-polarized output phase in channel L-R and R-L are still maintained in same state. It is indicated that the interior rotation angle ϑ would just produce decoupling influence in the co-polarized channels and would have no effect on the cross-polarized output fields. It should be noted that in this part, equation (S5) is established on the idealized assumption that the three patch layers are considered as perfectly independent wave plates. However, practically in this work, the three patch layers are metallic elements in the equivalent filter circuit as analyzed in Text. 1, meaning that the coupling between each

layer would unavoidably affect the output final phase. Therefore, the phase in four CP channels of each meta-atom for metadevices construction (as illustrated in Fig. 3b), are finally fixed with full wave simulation and optimization processing, which can effectively counteract the influence produced by metallic-layer coupling.

Moreover, as for the proposed meta-atom imposed with both interior and exterior rotation, the

equivalent Jones matrix is expressed as $T = \begin{bmatrix} T_{LL} & T_{LR} \\ T_{RL} & T_{RR} \end{bmatrix}$, which is further calculated based on T_I . It is

known that the exterior rotation provid geometric phase and the exterior rotation angle is set as θ . The derivation process can be described as follows:

$$\begin{aligned}
 T(\vartheta, \theta) &= M(\theta)^T \cdot T_I(\vartheta) \cdot M(\theta) \\
 &= \begin{bmatrix} \cos \theta & -\sin \theta \\ \sin \theta & \cos \theta \end{bmatrix} \cdot \begin{bmatrix} T_{LL} & T_{LR} \\ T_{RL} & T_{RR} \end{bmatrix} \cdot \begin{bmatrix} \cos \theta & \sin \theta \\ -\sin \theta & \cos \theta \end{bmatrix} \\
 &= \begin{bmatrix} T_{LL} \cdot \cos^2 \theta + T_{RR} \cdot \sin^2 \theta - (T_{ILR} + T_{IRL}) \cdot \sin \theta \cdot \cos \theta & (T_{RR} - T_{LL}) \cdot \sin \theta \cdot \cos \theta + T_{ILR} \cdot \cos^2 \theta - T_{IRL} \cdot \sin^2 \theta \\ (T_{RR} - T_{LL}) \cdot \sin \theta \cdot \cos \theta - T_{ILR} \cdot \sin^2 \theta + T_{IRL} \cdot \cos^2 \theta & T_{LL} \cdot \sin^2 \theta + T_{RR} \cdot \cos^2 \theta + (T_{ILR} + T_{IRL}) \cdot \sin \theta \cdot \cos \theta \end{bmatrix} \\
 &= \begin{bmatrix} A & B \\ C & D \end{bmatrix}
 \end{aligned} \tag{S6}$$

where $M(\theta) = \begin{bmatrix} \cos \theta & \sin \theta \\ -\sin \theta & \cos \theta \end{bmatrix}$ is the rotation matrix. Equation (S6) expresses the general form of

Jones matrix. Four elements A, B, C, D are independent from each other and represent for different transmission characteristics in the four channels, which are described as:

$$A = \frac{1}{2}(T_{LL} + T_{RR}) + \frac{1}{4}[(T_{LL} - T_{RR}) + i \cdot (T_{ILR} + T_{IRL})] \cdot e^{i2\theta} + \frac{1}{4}[(T_{LL} - T_{RR}) - i \cdot (T_{ILR} + T_{IRL})] \cdot e^{-i2\theta} \tag{S7a}$$

$$B = \frac{1}{2}(T_{ILR} - T_{IRL}) + \frac{1}{4}[(T_{ILR} + T_{IRL}) - i \cdot (T_{LL} - T_{RR})] \cdot e^{i2\theta} + \frac{1}{4}[(T_{ILR} + T_{IRL}) + i \cdot (T_{LL} - T_{RR})] \cdot e^{-i2\theta} \tag{S7b}$$

$$C = \frac{1}{2}(T_{IRL} - T_{ILR}) + \frac{1}{4}[(T_{ILR} + T_{IRL}) - i \cdot (T_{LL} - T_{RR})] \cdot e^{i2\theta} + \frac{1}{4}[(T_{ILR} + T_{IRL}) + i \cdot (T_{LL} - T_{RR})] \cdot e^{-i2\theta} \tag{S7c}$$

$$D = \frac{1}{2}(T_{LL} + T_{RR}) + \frac{1}{4}[(T_{RR} - T_{LL}) - i \cdot (T_{ILR} + T_{IRL})] \cdot e^{i2\theta} + \frac{1}{4}[(T_{RR} - T_{LL}) + i \cdot (T_{ILR} + T_{IRL})] \cdot e^{-i2\theta} \tag{S7d}$$

From equation (S5) we can note that the two cross-polarized phase responses are similar to each other when there is only interior rotation in the meta-atom structure, leading to $\phi_{ILR} = \phi_{IRL}$. Through substituting Equation (S3) into Equation (S6-S7), the meta-atom with both interior and exterior rotations can be equivalent to Jones matrix of meta-atom as described:

$$\begin{aligned}
 A(t_{0xx}, t_{0yy}, \vartheta, \theta) &= \frac{1}{8} [(t_{0xx} + t_{0yy})^3 + (t_{0xx} + t_{0yy}) \cdot (t_{0xx} - t_{0yy})^2 \cdot (2 \cos 2\vartheta + \cos 4\vartheta)] \\
 &\quad + \frac{i}{8} [(t_{0xx} + t_{0yy}) \cdot (t_{0xx} - t_{0yy})^2 \cdot (2 \sin 2\vartheta + \sin 4\vartheta)] \cos 2\theta \\
 &\quad + \frac{1}{8} [(t_{0xx} - t_{0yy})^3 + 2 \cdot (t_{0xx} + t_{0yy})^2 \cdot (t_{0xx} - t_{0yy}) \cdot \cos 2\vartheta] \sin 2\theta
 \end{aligned} \tag{S8a}$$

$$\begin{aligned}
 B(t_{0xx}, t_{0yy}, \vartheta, \theta) &= \frac{i}{8} [(t_{0xx} + t_{0yy}) \cdot (t_{0xx} - t_{0yy})^2 \cdot (2 \sin 2\vartheta + \sin 4\vartheta)] \cdot \sin 2\theta \\
 &\quad + \frac{1}{8} [(t_{0xx} - t_{0yy})^3 + 2 \cdot (t_{0xx} + t_{0yy})^2 \cdot (t_{0xx} - t_{0yy}) \cdot \cos 2\vartheta] \cdot \cos 2\theta
 \end{aligned} \tag{S8b}$$

$$\begin{aligned}
 C(t_{0xx}, t_{0yy}, \vartheta, \theta) &= \frac{i}{8} [(t_{0xx} + t_{0yy}) \cdot (t_{0xx} - t_{0yy})^2 \cdot (2 \sin 2\vartheta + \sin 4\vartheta)] \cdot \sin 2\theta \\
 &\quad + \frac{1}{8} [(t_{0xx} - t_{0yy})^3 + 2 \cdot (t_{0xx} + t_{0yy})^2 \cdot (t_{0xx} - t_{0yy}) \cdot \cos 2\vartheta] \cdot \cos 2\theta
 \end{aligned} \tag{S8c}$$

$$\begin{aligned}
 D(t_{0xx}, t_{0yy}, \vartheta, \theta) &= \frac{1}{8} [(t_{0xx} + t_{0yy})^3 + (t_{0xx} + t_{0yy}) \cdot (t_{0xx} - t_{0yy})^2 \cdot (2 \cos 2\vartheta + \cos 4\vartheta)] \\
 &\quad - \frac{i}{8} [(t_{0xx} + t_{0yy}) \cdot (t_{0xx} - t_{0yy})^2 \cdot (2 \sin 2\vartheta + \sin 4\vartheta)] \cos 2\theta \\
 &\quad + \frac{1}{8} [(t_{0xx} - t_{0yy})^3 + 2 \cdot (t_{0xx} + t_{0yy})^2 \cdot (t_{0xx} - t_{0yy}) \cdot \cos 2\vartheta] \sin 2\theta
 \end{aligned} \tag{S8d}$$

where t_{0xx} and t_{0yy} represent the linear transmission coefficients of birefringent single-layer patch, ϑ is the interior rotation angle provided by three independent patch layers, and θ shows the exterior angle introduced by rotating the whole meta-atom with all three patch layers.

Comments: I have another concern about the fabrication of the multi-layer metasurface. This work demonstrates it in microwave so the antenna is relatively large. But when applying it to make optical metasurfaces, manufacturing these layers and the alignment across multiple layers of optical metasurfaces can be a serious hurdle, as far as I know. This would largely limit the applicability of the proposed method.

Response: The reviewer is right that the devices are fabricated for microwave frequencies and are relatively large and practical in terms of experimental realization. The proposed metasurface in our paper contains $25 * 25$ meta-atoms along x - and y -directions. The dimension size of the metasurface is $7.3\lambda_0 * 7.3\lambda_0$ (220 mm * 220 mm, where $\lambda_0 = 30$ mm). For the meta-atom, the size is about $0.3\lambda_0 * 0.3\lambda_0$, which indicates that the antenna size is not large with respect to working wavelength. The realization of similar device that could operate at optical frequencies is yet certainly a challenge but supposing an operation wavelength of $1\mu\text{m}$, 300nm features have been demonstrated and could be realized. But we

agree with the referee that efforts in achieving operational metasurfaces at optical wavelength is generally needed, not only for the nanostructures proposed in our paper.

As for the multi-layer metasurface manufacturing, there already exists several fabricated samples in optical region. For example, multiple metallic layers meta-structure is fabricated in *10.1038/ncomms1877* (Fig. R1a), double-layer meta-atom is proposed to demonstrate asymmetric transmission in *10.1103/PhysRevLett.104.253902* (R1b), metasurface with multiple Au layers is fabricated to realize light asymmetric transmission in *10.1103/PhysRevLett.113.023902* (Fig. R1c). With further development, the optical fabrication technology will be mature enough to produce more complicated and precise optical structures.

On the other hand, the proposed metasurfaces in this paper are only demonstrations for the verification of synthesis of chirality-assisted phase, geometric phase and propagation phase, aiming for full phase manipulation of four output channels. In the microwave region, the technology of printed circuit board is available for the fabrication of the multi-layered structures, thus the specific structure of meta-atom in this work is adopted to provide the required three kinds of phase responses, but this doesn't mean the multi-layered meta-atom structure is necessary for similar effects in optical region. Other metallic or dielectric nano-structures can be appropriate for the fabrication of optical metasurfaces, as long as the three phase modulation degrees (propagation phase, geometric phase and chirality-assisted phase) can be simultaneously satisfied by tailoring some specific parameters of meta-atom. Our objective is to provide a general scheme for phase-manipulation of all four transmission channels simultaneously, and the proposed meta-atom structure is just a demonstration sample in microwave region. Similarly, most of the early papers on metamaterials have been demonstrated at microwave wavelengths, which triggered intense interest and motivation towards optical wavelengths.

This scheme extension is illustrated in main text, which is added at the end of part 3 in the main text and the corresponding description is in red.

Added or Revised Text:

Here it should be noticed that the phase manipulation to achieve distinct responses in all four transmission CP channels is a general method, which is not limited by operating wavelength. Although the concept is demonstrated in microwave region in this work, the method can be extended and applied in other wavelength regimes, as long as the appropriate meta-atom structure can be designed to

simultaneously implement the three phase responses (propagation phase, chirality-assisted phase and geometric phase) by tailoring corresponding parameters.

Fig. R1 Fabrication methods of optical metasurfaces with multiple metallic layers.

Comments: Minor comments:

I think the authors need to proof read their draft more carefully. There are quite a few typos or grammar issues. here are a few examples (among others):

- Line 81: meatsurfaces -> metasurfaces

Response: The authors are grateful for these detailed comments. “meatsurfaces” in previous Line 81 has been revised into “metasurfaces”, which is highlighted with red color in revised manuscript.

Comments: - Line 114: to decouple inherent consistent -> to decouple inherent consistency?

Response: “consistent” in previous Line 114 has been revised into “consistency” and highlighted in red in revised manuscript.

Comments: - Line 58 of supplementary: the subscripts of α should be 1, 2, 3.

Response: The subscript of α has been changed into 1, 2, 3 and highlighted in red in revised Supporting Information.

Comments: - In eq. (S3), I believe the bra and ket in each line should be switched.

Response: The reviewer is right, we have switched bra and ket in Eq. (S2) in revised Supporting Information, which is corresponding to Eq. (S3) in previous manuscript.

Added or Revised Text:

$$\begin{aligned}
 T_{cas} = & \frac{1}{8}(t_{0xx} + t_{0yy})^3 \cdot I \\
 & + \frac{1}{8}[(t_{0xx} + t_{0yy}) \cdot (t_{0xx} - t_{0yy})^2 \cdot (2e^{i2\theta} + e^{i4\theta})] \cdot |L\rangle\langle L| \\
 & + \frac{1}{8}[(t_{0xx} + t_{0yy})(t_{0xx} - t_{0yy})^2 \cdot (2e^{-i2\theta} + e^{-i4\theta})] \cdot |R\rangle\langle R| \\
 & + \frac{1}{8}[(t_{0xx} - t_{0yy})^3 + (t_{0xx} + t_{0yy})^2 \cdot (t_{0xx} - t_{0yy}) \cdot (1 + e^{-i2\theta} + e^{i2\theta})] \cdot |L\rangle\langle R| \\
 & + \frac{1}{8}[(t_{0xx} - t_{0yy})^3 + (t_{0xx} + t_{0yy})^2 \cdot (t_{0xx} - t_{0yy}) \cdot (1 + e^{i2\theta} + e^{-i2\theta})] \cdot |R\rangle\langle L|
 \end{aligned}$$

Comments: - Line 64 of supplementary: "amplitude of co-polarized and cross-polarized are" not grammatically correct.

Response: We agree with the reviewer about this comment. We have re-clarified the derivation process, and the statement about "amplitude of co-polarized and cross-polarized" has been revised.

Comments: - Line 77 of supplementary: "independent perfect wave plates" you mean "perfectly independent wave plates"?

Response: The authors thanks the reviewer about this detailed comment. The statement of "independent perfect wave plates" in original Line 77 has been changed into "perfectly independent ideal wave plates" in the revised Supporting Information.

Reviewer #2 (Remarks to the Author):

Comments: While past work has demonstrated arbitrary symmetry and unitary Jones matrices operating in transmission mode, the authors demonstrate extended functionality by including chirality to construct Jones matrices that are not symmetric. They show that by controlling the propagation phase, geometric phase, and chirality, the phase of four distinct channels of circularly polarization transmitting through the device may be controlled. The results and approach are non-obvious and impressive, and of great interest to the community. The presentation of the results is excellent, though the English could use some work. The explanation lacks clarity in places, has a few inaccuracies, and is burdened by poor notation, meaning the pedagogical worth of the paper is lacking. Assuming these issues and the following two major corrections are addressed, I am happy to recommend this work for publication in Nature Communications. I detail major and minor issues below.

Response: The authors are grateful for the reviewer's positive feedback and recommendation.

Comments: Major issues whose resolution I consider necessary for publication:

1. The assumptions on the form of the (4x4) scattering matrices of each meta-atom being achieved here should be stated explicitly. It appears from their description that the authors assume a reciprocal, lossless, matched system. In this case, the scattering matrix is symmetric and unitary, and there is negligible reflection (impedance matched). If this is true, two statements the authors make are demonstrably false. If this is not true, the authors should clarify and address how this resolves the following problems:

(a) The authors claim that the four transmission coefficients in the Jones matrix are totally independent. This is not true if the system is reciprocal (passive elements only), lossless (no material losses), and matched (i.e. no reflection losses either). It is easy to show that $\text{abs}(t_{xx}) = \text{abs}(t_{yy})$ and $\text{abs}(t_{xy}) = \text{abs}(t_{yx})$ in this case. The phases of each component, however, are indeed independent in principle (the complex Jones matrix need not be symmetric). In transmission mode, reciprocity constrains the phase of t_{xy} from side 1 to 2 to be the same as t_{yx} from side 2 to 1, but does not constrain the phases of t_{xy} and t_{yx} when incident from the same side. This inaccuracy does not change the authors' results, since they focus on phase manipulation and choose all of these amplitudes to be equal to each other anyways. Therefore, this is easily fixed by clarifying the relevant passages to specify that the phases are independent.

Response: The authors thanks the reviewer for this valuable comment. We agree the reviewer that for the complex Jones matrix, four transmission coefficients are not totally independent, and the aim of this paper is to achieve independent phase-modulation of all 4 channels in transmission. There are four parameter degrees of freedom to decouple the inherent relevance between 4 transmission coefficients: p_x, p_y (width and length of the rectangular patch) for propagation phase modulation, ϑ (interior rotation angle) for chirality-assisted phase modulation, and θ (exterior rotation angle) for geometric phase modulation. Since the proposed metasurface constructed without any active component and the quasi no material loss in microwave region, the system is assumed to be passive and lossless. Detailed analysis about the meta-atom and the equivalent network are as follows.

The scattering matrix with different propagation directions of proposed transmission-type meta-atom can be separately described as:

$$S_{12} = \begin{bmatrix} S_{12}^{xx} & S_{12}^{xy} \\ S_{12}^{yx} & S_{12}^{yy} \end{bmatrix}, S_{21} = \begin{bmatrix} S_{21}^{xx} & S_{21}^{xy} \\ S_{21}^{yx} & S_{21}^{yy} \end{bmatrix} \quad (R3)$$

where the first (second) subscript presents for the region of incident (scattered) field, and the first (second) superscript shows the polarization state of incident (scattered) wave. As for the proposed transmission-mode meta-atom, the transmission characteristics of its equivalent network are then discussed.

1. Reciprocity. Since the meta-atom design proposed in this paper is symmetric in structure and totally passive (without any active component), all of the proposed meta-atoms are reciprocal. It means that there exist $S_{12}^{xy} = S_{21}^{yx}$ and $S_{12}^{yx} = S_{21}^{xy}$ in the general scattering matrix. Additionally, the reciprocity is verified by simulations as shown in Fig. R2, where the scattering responses of transmission coefficients are exhibited. Figs. R2a and R2b show the amplitude and phase responses of meta-atom with $p_x = 4.8$ mm, $p_y = 5.7$ mm, $\vartheta = 0$ and $\theta = 0$, Figs. R2c and R2d exhibit the amplitude and phase responses of meta-atom with $p_x = 4.8$ mm, $p_y = 5.7$ mm, $\vartheta = 30^\circ$ and $\theta = 0$, Figs. R2e and R2f present the amplitude and phase responses of meta-atom with $p_x = 4.8$ mm, $p_y = 5.7$ mm, $\vartheta = 30^\circ$ and $\theta = 30^\circ$. It can be seen that the transmission curves S_{12}^{xy} and S_{21}^{yx} of all these three meta-atoms are in agreement, while S_{12}^{yx} and S_{21}^{xy} are also equal, which successfully verify the reciprocity of proposed structure.

2. Lossless. In microwave region, metal can be regarded as perfect electric conductor and the loss tangent of dielectric materials is usually in the order of 10^{-3} . Thus, the system can be supposed as a lossless system in the analyses and simulations. Practically, if the system is proposed with lossy media,

for example metallic meta-atoms in optical region, the equivalent scattering matrix would not be unity and the efficiency of the system would be further affected and reduced. Since our scheme only works on the phase modulation, the change of amplitude would not affect the application of the scheme and it is still effective.

3. Matching. The theoretical model is supposed to be matched to free space, which is convenient for calculating the phase-modulation process. However, since there is inevitable reflection from the metasurfaces (or meta-atoms), the practical system would not be matched. During the construction of metasurfaces, the phase responses of meta-atoms have to approach discrete spatial phase distribution, for example required for deflection or OAM generation. To achieve the phase response as accurate as possible, there would be some optimization process of the meta-atom's phase response and the amplitude response would inevitably be sacrificed. On the other hand, the interior and exterior rotation in meta-atom structure would also introduce mismatching in the system, resulting in the increase of reflection components. As it can be observed from Figs. R2a, R2c and R2e, when the meta-atom is imposed with the interior and exterior rotations, the amplitude of S^{xx} and S^{yy} are gradually decreased. So, equivalent network of the meta-atom is considered mismatched to free space.

In fact, in our proposed scheme for full transmission phase manipulation, we just consider one propagation direction (from port 1 to port 2) of the electromagnetic wave. As mentioned by the reviewer, the four transmission coefficients are not totally independent to each other. Indeed, there exists inherent consistency between them. When the meta-atom is set with certain rectangular patch dimensions and no rotation factors (fixed p_x and p_y , $\vartheta = 0$ and $\theta = 0$), there exists $S_{LL} = S_{RR}$, $S_{LR} = S_{RL}$ in the scattering matrix attributed to the birefringent characteristics. Therefore, with the propagation phase manipulation, distinct phase responses can be produced in channels S_{LL} ($= S_{RR}$) and S_{LR} ($= S_{RL}$) firstly. Then the interior rotation angle is introduced to decouple the consistency between S_{LL} and S_{RR} , while keeping S_{LR} and S_{RL} unchanged. Finally, the exterior rotation angle can further realize distinct responses of S_{LR} and S_{RL} , while producing no effect on S_{LL} and S_{RR} .

Based on this modulation scheme and the spatial phase distributions of required wavefronts, four transmission coefficients can be separated and the independent functionalities can be imposed in four channels. The clarification of the system characteristics being important for manipulation light performances, we have revised the inaccuracy of previous statements and added the corresponding analyzation in revised manuscript.

Fig. R2 Simulated scattering responses of three proposed meta-atoms with different parameter dimensions, where (a), (c), (e) show the amplitude responses and (b), (d), (f) exhibit the phase profiles.

Added or Revised Text:

i). In main text

The equivalent metasurface system is supposed to be passive, lossless, matched and reciprocal.

ii). In main text

Here a passive, reciprocal transmission-type meta-atom is applied to implement all these three phase modulation schemes.

iii). In Supporting Information

Furthermore, the basic electromagnetic characteristics of this proposed meta-atom and its equivalent network is discussed in this part.

1. Reciprocity. Since the meta-atom design proposed in this paper is symmetric in structure and totally passive (without any active component), all of the proposed meta-atoms are reciprocal. It means that

there exist $S_{12}^{xy} = S_{21}^{yx}$, $S_{12}^{yx} = S_{21}^{xy}$ in the general scattering matrix, where the first (second) subscript presents for the region of incident (scattered) field, and the first (second) superscript shows the polarization state of incident (scattered) wave. Additionally, the reciprocity is verified by simulations as shown in Fig. S2, where the scattering responses of transmission coefficients are exhibited. Figs. S2(a) and S2(b) show the amplitude and phase responses of meta-atom with $p_x = 4.8$ mm, $p_y = 5.7$ mm, $\vartheta = 0$ and $\theta = 0$, Fig. S2(c) and S2(d) exhibit the amplitude and phase responses of meta-atom with $p_x = 4.8$ mm, $p_y = 5.7$ mm, $\vartheta = 30^\circ$ and $\theta = 0$, Figs. S2(e) and S2(f) present the amplitude and phase responses of meta-atom with $p_x = 4.8$ mm, $p_y = 5.7$ mm, $\vartheta = 30^\circ$ and $\theta = 30^\circ$. It can be seen that the transmission curves S_{12}^{xy} and S_{21}^{yx} of all these three meta-atoms are in agreement, while S_{12}^{yx} and S_{21}^{xy} are also equal, which successfully verify the reciprocity of proposed structure. In fact, in our proposed scheme for full transmission phase manipulation, we just consider one propagation direction (from port 1 to port 2) of the electromagnetic wave.

2. Lossless. In microwave region, metal can be regarded as perfect electric conductor and the loss tangent of dielectric materials is usually in the order of 10^{-3} . Thus, the system can be supposed as a lossless system in the analyses and simulations. Practically, if the system is proposed with lossy media, the equivalent scattering matrix would not be unitary and the efficiency of the system would be further affected and reduced. Since our scheme only works on the phase modulation, the change of amplitude would not affect the application of the scheme and it is still effective.

3. Matching. The ideal model is supposed to be matched to free space, which is convenient for calculating the phase-modulation process. However, since there is inevitable reflection within the design process of metasurface (or meta-atom) structures, the practical system would be mismatched. During the construction of metasurfaces, the phase responses of meta-atoms have to approach discrete spatial phase distribution, for example required for deflection or OAM generation. To achieve the phase response as accurate as possible, there would be some optimization process of the meta-atom's phase response and the amplitude response would inevitably be sacrificed. On the other hand, the interior and exterior rotation in meta-atom structure would also introduce mismatching in the system, resulting in the increase of reflection components. As it can be observed from Figs. S2a, S2c and S2e, when the meta-atom is imposed with the interior and exterior rotations, the amplitude of S^{xx} and S^{yy} are gradually decreased. So, equivalent network of the meta-atom is considered mismatched to free space.

Fig. S2 Simulated scattering responses of three proposed meta-atoms with different parameter dimensions, where **a, c, e** show the amplitude responses and **b, d, f** exhibit the phase profiles.

Comments: (b) The authors claim that the feat can be achieved in Reflection mode as well (where the matched assumption is omitted, of course, and presumably a mirror eliminates the transmission-side ports). This is not true if the system is reciprocal, because $rx_y = ry_x$ in this case. In other words, the complex Jones matrix is symmetric in reflection mode by reciprocity. That is, both the amplitude and phase of the off-diagonal components of the reflected Jones matrix are identical. There simply aren't four independent phases to control in a reciprocal mirror with one diffraction order. Again, this does not change the authors' results for transmission mode. However, it is a basic inaccuracy that should be remedied before publication.

Response: The authors are really grateful for this pertinent comment. As for a reflection-type metasurface, the metallic ground would limit the port number of the system, the input and output ends

are set on the same side of metasurface. For a reciprocal, lossless and passive meta-atom system, the off-diagonal elements in reflection matrix, which represent for the reflectance of two co-polarization channels, would be theoretically identical, as mentioned in the comment. The amplitude and phase responses in off-diagonal channels would be inherent consistent. Therefore, the chirality-assisted phase cannot be introduced in the co-polarization channels with reflection mode, indicating that these two channels cannot be independently modulated through phase tuning. Indeed, the working principle of reflection-mode metasurface is actually different from that of the transmission one. We have deleted this inaccurate statement in the revised manuscript, and made the paper more rigorous. We thank the reviewer for his/her careful reading and deep physical analysis, which helped us not shortcutting the reflection case.

Comments: The relationship between dependent and independent degrees of freedom is not made completely clear, even after reading the main text and supplementary several times. For a reciprocal, lossless, matched system, there are 5 degrees of freedom in the transmitted Jones matrix: four phases, and one relative amplitude (ratio between a diagonal component and an off-diagonal component). The authors vary four geometric degrees of freedom: p_x , p_y , internal angle, and external angle. By using one of these degrees of freedom, one may easily set the amplitude of diagonal and off-diagonal components to be equal. It is implied that the degree of birefringence is fixed to set the amplitude to be equal, and the uninquisitive reader is thereby led to believe that this is achievable for any choice of the four phases. But this constraint uses up a degree of freedom (e.g. it fixes p_y for a choice of p_x and internal angle). This means there are three degrees of freedom remaining to control the desired four phases (in each the LL,RR,LR,RL channels), which is not feasible even in principle.

Alternatively, the constraint on equal amplitude may be relaxed so that the four phases may be controlled independently. In supplementary section S5, the authors briefly discuss how the transmission amplitude of each component is not, in fact, perfectly equal. The authors explain this by the fact that the underlying phase-control mechanism is a resonance, which does not have constant amplitude as a function of p_x and p_y . However, this complication notwithstanding, it is not discussed in the manuscript whether the amplitudes can be equal even in principle (e.g. by using dielectric structures which impart phase in a non-resonant manner). The simple degrees-of-freedom argument above suggests that it is not possible without a fifth degree of freedom.

This limitation should be clarified explicitly, and the impact on achievable devices should be discussed. The fact that the amplitude of each channel is not independent of the chosen phases is highly relevant to experts in the field. Great effort was made in the initial metasurface works (e.g., the classic Ref [8]) to maintain constant amplitude, so that the phase response alone controls the local diffraction efficiencies across the device. This is a key assumption behind the Generalized Snell's Law (Ref [8]), used by the authors in Equation (3), wherein a single diffraction order is selected for locally. A direct result of this amplitude limitation in the authors' case is that the diffraction efficiencies into the desired channels may not all simultaneously be 50% (the ideal case for the authors) for arbitrary phase functions even in principle. The impact on diffraction efficiencies due to amplitude variations may partially be compensated, as briefly mentioned by the authors in the supplementary section S5, but not perfectly.

I personally do not think this limitation of the design is enough to warrant rejection from Nature Communications, though it means the claims of the paper need to be clarified. In my view, the results speak for themselves: reasonable control over four output channels is achievable, which is in my view a significant advance to the field. It is well-known that phase is more important than amplitude in diffractive devices, so the variance of amplitude as a cost of control of 4 phase is a worthwhile, if somewhat limited, achievement. However, the implied assumption behind the vast majority of metasurface works is that the efficiencies could be ideal in principle, and are limited in practice by various imperfections in the control of optical responses and fabrication errors. Future work solving these practical and engineering imperfections are then immediately inheritable. Here, on the other hand, the efficiencies are limited by construction: it is a limitation in the design approach itself. This difference is a crucial to spell out, and a paper that hides it rather than clarifies it should be rejected. I think it is appropriate to say that future work can add that 5th degree of freedom to fully control the light, while the current manuscript represents a step to that final goal.

Response: The authors are grateful for the valuable comments. As mentioned by the reviewer, based on the current mechanism in this paper, the amplitude response of transmission coefficients is indeed beyond the scope of manipulation, and the independent phase-modulation of all 4 transmission channels is the key aim of this paper, and the inappropriate statement of ideal amplitude manipulation has been revised.

In this paper, we applied four parameter degrees of freedom to manipulate all phase patterns of four transmission coefficients independently and simultaneously. The dimension sizes of patch layer (p_x and p_y) can originally decide the phase profile of diagonal and off-diagonal elements, achieving independent φ_{LL} ($= \varphi_{RR}$) and φ_{LR} ($= \varphi_{RL}$). Then the interior rotation angle ϑ can produce chiral responses in structure, which can decouple the phase response between φ_{LL} and φ_{RR} . After that, the exterior angle θ is applied to impose equal and opposite phase interruption in two cross-polarized components, resulting in the distinct response of φ_{LR} and φ_{RL} . These four degrees of freedom for decoupling the inherent consistency among four phase responses are the primary condition within the whole phase modulation scheme.

It is noted that when there is no interior or exterior rotation ($\vartheta = \theta = 0$), the amplitude ratio between the diagonal and off-diagonal components can be determined by the phase difference between the linearly polarized phase responses along the fast- and slow-axes of birefringent meta-atom, which is guaranteed by p_x and p_y . That is to say, if the amplitudes are completely modulated by this linear-phase-difference factor, p_x and p_y would be fixed and limited by this condition. This means these two parameter degrees of freedom to define original phase pattern of diagonal and off-diagonal components would be invalid. Considering the primary condition for our proposed synthesis of three kinds of phase, phase modulation is more important than amplitude manipulation. Thus the referee is totally right, the amplitude is indeed out of the first consideration and sacrificed during the optimization process for guaranteeing the phase requirements.

In former version of the manuscript, we derived this amplitude manipulation condition for distributing equal output energy in diagonal and off-diagonal components, in order to set an ideal initial design criterion for structure optimization process. However, in fact, only a few meta-atoms can satisfy the amplitude criterion after optimization for required phase profiles, as shown in Fig. S3. For a clear illustration of the limitation between the number of degrees of freedom and amplitude control, we have added this discussion on amplitude limitation in the revised main text and supporting information, and highlighted with red color.

In the future, we will focus on the extra degrees of freedom (e.g. material losses, active components, even time) for independent amplitude modulation based on current works.

Added or Revised Text:

i). In main text:

It is noted that, in this scheme we applied four parameter degrees of freedom (p_x , p_y , ϑ and θ) to independently manipulate four phase patterns of transmission channels (φ_{LL} , φ_{LR} , φ_{RL} , and φ_{RR}). However, within this phase-modulation process, the amplitude responses are not considered and the efficiency is partly sacrificed due to the particularity of the proposed meta-atom structure. This limitation is detailed in Supporting Information. The exploration of an extra degree of freedom (e.g. material losses, active components, even time) for independent amplitude modulation is therefore desired in future works.

ii). In main text:

The proportion between the sum of energy distributed in both L-L and L-R channels to that in R-L and R-R channels is approximately 1 : 3, corresponding to the ratio between LHCP and RHCP component at the input end with RHEP state. Fig. 4c displays the far-field intensity under LP wave illumination, where the far-field lobes in L-L and L-R channels exhibit nearly equal amplitude to that in R-L and R-R channels. When the input state is switched to LHEP, the proportion of output sum energy in both L-L and L-R channels to that in R-L, R-R channels is changed to 3 : 1 and displayed in Fig. 4d, as reversely symmetric situation to that in RHEP results.

iii). In Supporting Information:

In this part, the amplitude response of meta-atoms is discussed. Here, we apply four parameter degrees of freedom to manipulate all phase patterns of four transmission coefficients independently and simultaneously. The dimension sizes of patch layer (p_x and p_y) can originally set the phase profile of diagonal and off-diagonal elements, achieving independent φ_{LL} ($= \varphi_{RR}$) and φ_{LR} ($= \varphi_{RL}$). Then the interior angle ϑ can produce chiral responses in structure, which can decouple the phase response between φ_{LL} and φ_{RR} . After that, the exterior angle θ is applied to impose equal and opposite phase interruption in two cross-polarized components, resulting in the distinct response of φ_{LR} and φ_{RL} . These four degrees of freedom for decoupling the inherent consistency between four phase responses are the primary conditions within the whole phase modulation scheme.

It is noted that when there is no interior or exterior rotation ($\vartheta = \theta = 0$), the amplitude ratio between the diagonal and off-diagonal components can be determined by the phase difference between the linearly polarized phase responses along the fast- and slow-axes of birefringent meta-atom as illustrated in Equation (S4), which is guaranteed by p_x and p_y . Therefore, if the amplitudes are modulated by this linear-phase-difference factor, p_x and p_y would be fixed and limited by this condition. This means these

two parameter degrees of freedom to define original phase pattern of diagonal and off-diagonal components would be invalid. Considering the primary condition for our proposed synthesis of three kinds of phase resources, phase modulation is more important than amplitude manipulation. Thus, the amplitude is indeed not considered and partly sacrificed during the optimization process to guarantee the phase requirements.

As mentioned in the main text, there are 25 meta-atoms collected to provide the required phase gradients in all four channels along x -direction. The calculated and simulated phase profiles of all meta-atoms are exhibited in Fig. 2b in the main text, while the corresponding simulated output amplitudes of CP conversion channels L-L, L-R, R-L, and R-R are illustrated in Fig. S4. It can be further observed that the amplitude responses of meta-atoms are not homogeneous, which is sacrificed to the phase limitations as explained above. We have to say that the non-uniform amplitudes would have some influence in energy distribution, such as the intensity mappings shown in Fig. 5a and 5b, where the doughnut-shape energy rings of vortex beams exhibit some heterogeneity. However, these unequal amplitude responses would not affect the phase distribution, meaning that the proposed phase-modulation scheme would not be limited by the sacrificed amplitude responses.

For future works, an extra degree of freedom needs to be explored to modulate the amplitude profiles so as to achieve independent manipulation of phase and energy in the four CP transmission channels.

Comments: Minor issues that I suggest resolving to improve the manuscript:

1. In Figure 5, the use of a black background to organize the data is inappropriate if black is included in the colormap representing the data. It is sometimes difficult to tell where the data ends and where the background starts. A drawn border indicating these edges would also fix this issue without changing the background black color.

Response: We agree with the reviewer, and the background color of Figure 5 has been changed. New Figure 5 has been added in the revised manuscript.

Added or Revised Text: Fig. 5 with changed background is shown as,

Comments: The notation (e.g. $E_{(LHCP,inc)^{(out-co)}}$) is needlessly cumbersome, and hinders comprehension. Why not use $E_{LL}, E_{RL}, E_{LR}, E_{RR}$ instead? This immediately tells the reader all the relevant information. It is readily apparent that E_{RL}, E_{LR} are cross-polarized components while E_{LL}, E_{RR} are co-polarized components.

Response: Following the comment, all the complicate notations are changed to give a simple and clear expression.

Added or Revised Text:

When the proposed metasurface is illuminated by LHCP incident wave $|\bar{L}\rangle = \begin{bmatrix} 1 \\ i \end{bmatrix}$ and RHCP incident

wave $|\bar{R}\rangle = \begin{bmatrix} 1 \\ -i \end{bmatrix}$ respectively, the output electric fields can be expressed as:

$$\begin{aligned}
 \vec{E}_{out}^{L, in} &= T \cdot |L\rangle \\
 &= (t_{propa}^{co} + t_{chiral}^{co}) \cdot |L\rangle + (t_{propa}^{cross} - t_{chiral}^{cross}) \cdot e^{i\phi_{geo}} \cdot |R\rangle \\
 &= \vec{E}_{LL} + \vec{E}_{LR}
 \end{aligned} \tag{2a}$$

$$\begin{aligned}
 \vec{E}_{out}^{R,in} &= T \cdot |R\rangle \\
 &= (t_{propa}^{co} - t_{chiral}^{co}) \cdot |R\rangle + (t_{propa}^{cross} + t_{chiral}^{cross}) \cdot e^{-i\phi_{geo}} \cdot |L\rangle \\
 &= \vec{E}_{RR} + \vec{E}_{RL}
 \end{aligned} \tag{2b}$$

It can be observed from Equation (2) that under the orthogonal CP illuminations, the metasurface can produce four different output fields, including two co-polarized output components \vec{E}_{LL} and \vec{E}_{RR} , and two cross-polarized components \vec{E}_{LR} and \vec{E}_{RL} . Based on the analyses above, four CP transmission coefficients can be independently phase-modulated, where the inherent consistencies between four output phase patterns ϕ_{LL} , ϕ_{LR} , ϕ_{RL} , and ϕ_{RR} are independently decoupled by propagation phase, chirality-assisted phase and geometric phase, respectively.

Comments: The matrices T1 and T2, defined in Equation (1), are never again referred to.

Response: Equation (1) has been replaced by the simple transmission coefficients with general formalism to exhibit a clear illustration of complex Jones matrix in the revised manuscript.

Added or Revised Text: Equation (1) is changed as

$$t_{LL} = \frac{1}{2}[(t_{xx} + t_{yy}) + i \cdot (t_{xy} - t_{yx})] \tag{1a}$$

$$t_{LR} = \frac{1}{2}[(t_{xx} - t_{yy}) - i \cdot (t_{xy} + t_{yx})] \cdot e^{i2\theta} \tag{1b}$$

$$t_{RL} = \frac{1}{2}[(t_{xx} - t_{yy}) + i \cdot (t_{xy} + t_{yx})] \cdot e^{-i2\theta} \tag{1c}$$

$$t_{RR} = \frac{1}{2}[(t_{xx} + t_{yy}) - i \cdot (t_{xy} - t_{yx})] \tag{1d}$$

Comments: Related to point 2 in the “major issues” list above, the co and cross phase terms in Equation S5 appear as separate, while the discussion around Equation S2 specifically mentions that they are the same. Why do they appear separate in Equation S5?

Response: The previous Equation S5 gives the phase terms of the meta-atom structure with specific interior rotation as illustrated and without exterior rotation, which includes 3 cascading metallic patch layers. Meanwhile, the co- and cross-polarized phases in previous Equation S2 represent the phase term of one metallic patch layer (not the whole structure), which is considered as the ideal one-layer birefringent wave plate. In order to illustrate the analysis process of the cascaded model of meta-atom,

we have deleted Equation S2 and revised the derivation steps, which is illustrated in red color in Supporting Information.

Comments: Relatedly, a closed-form relating the phases of $E_{LL}, E_{RL}, E_{LR}, E_{RR}$ to the phases of t_{xx} and t_{yy} , and the internal and external angle, would be highly valuable. Derivation of such a relationship would also show the amplitude variation discussed in point 2 above.

Response: The authors thanks for the valuable comments. In this work, the chirality-assisted phase responses are provided by the interior rotation angle of helical meta-atom structure. In order to show the influence introduced by interior rotation, we considered the meta-atom as a cascaded model with three similar rectangular ideal wave plates. According to the particularity of this meta-atom model, a closed form relating the phases of $E_{LL}, E_{LR}, E_{RL}, E_{RR}$ to the phases of t_{xx} and t_{yy} (the linear transmission coefficients of each *single* birefringent metallic layer), and the internal and external angle has been established, as shown in Equation (S8) in revised Supporting Information.

Added or Revised Text: revised Equation (S8) is shown,

$$\begin{aligned}
 A(t_{0xx}, t_{0yy}, \vartheta, \theta) &= \frac{1}{8}[(t_{0xx} + t_{0yy})^3 + (t_{0xx} + t_{0yy}) \cdot (t_{0xx} - t_{0yy})^2 \cdot (2 \cos 2\vartheta + \cos 4\vartheta)] \\
 &\quad + \frac{i}{8}[(t_{0xx} + t_{0yy}) \cdot (t_{0xx} - t_{0yy})^2 \cdot (2 \sin 2\vartheta + \sin 4\vartheta)] \cos 2\theta \\
 &\quad + \frac{1}{8}[(t_{0xx} - t_{0yy})^3 + 2 \cdot (t_{0xx} + t_{0yy})^2 \cdot (t_{0xx} - t_{0yy}) \cdot \cos 2\vartheta] \sin 2\theta
 \end{aligned} \tag{S8a}$$

$$\begin{aligned}
 B(t_{0xx}, t_{0yy}, \vartheta, \theta) &= \frac{i}{8}[(t_{0xx} + t_{0yy}) \cdot (t_{0xx} - t_{0yy})^2 \cdot (2 \sin 2\vartheta + \sin 4\vartheta)] \cdot \sin 2\theta \\
 &\quad + \frac{1}{8}[(t_{0xx} - t_{0yy})^3 + 2 \cdot (t_{0xx} + t_{0yy})^2 \cdot (t_{0xx} - t_{0yy}) \cdot \cos 2\vartheta] \cdot \cos 2\theta
 \end{aligned} \tag{S8b}$$

$$\begin{aligned}
 C(t_{0xx}, t_{0yy}, \vartheta, \theta) &= \frac{i}{8}[(t_{0xx} + t_{0yy}) \cdot (t_{0xx} - t_{0yy})^2 \cdot (2 \sin 2\vartheta + \sin 4\vartheta)] \cdot \sin 2\theta \\
 &\quad + \frac{1}{8}[(t_{0xx} - t_{0yy})^3 + 2 \cdot (t_{0xx} + t_{0yy})^2 \cdot (t_{0xx} - t_{0yy}) \cdot \cos 2\vartheta] \cdot \cos 2\theta
 \end{aligned} \tag{S8c}$$

$$\begin{aligned}
 D(t_{0xx}, t_{0yy}, \vartheta, \theta) &= \frac{1}{8}[(t_{0xx} + t_{0yy})^3 + (t_{0xx} + t_{0yy}) \cdot (t_{0xx} - t_{0yy})^2 \cdot (2 \cos 2\vartheta + \cos 4\vartheta)] \\
 &\quad - \frac{i}{8}[(t_{0xx} + t_{0yy}) \cdot (t_{0xx} - t_{0yy})^2 \cdot (2 \sin 2\vartheta + \sin 4\vartheta)] \cos 2\theta \\
 &\quad + \frac{1}{8}[(t_{0xx} - t_{0yy})^3 + 2 \cdot (t_{0xx} + t_{0yy})^2 \cdot (t_{0xx} - t_{0yy}) \cdot \cos 2\vartheta] \sin 2\theta
 \end{aligned} \tag{S8d}$$

where t_{0xx} and t_{0yy} represent the linear transmission coefficients of birefringent single-layer patch, ϑ is the interior rotation angle provided by three independent patch layers, and θ shows the exterior angle

introduced by rotating the whole meta-atom with all three patch layers.

Comments: Bold typeface appears to be used for both matrices and the elements of matrices, which is non-standard.

Response: The typeface of all matrices has been revised.

Comments: In Fig 5b the data is misaligned in the vertical direction.

Response: The data of Fig. 5b has been revised to be aligned in the vertical direction.

Added or Revised Text: the revised Fig. 5 with aligned data is shown as in response part to minor comment 1.

Comments: The title's phrase of "full wavefront modulation" is non-specific and not accurate given the 5th remaining degree of freedom being uncontrolled.

Response: The author thanks for this valuable comment, and the title is changed to "Independent phase modulation for quadruplex polarization channels enabled by chirality-assisted geometric-phase metasurfaces".

REVIEWERS' COMMENTS:

Reviewer #1 (Remarks to the Author):

Overall, I'm satisfied by this revision. The paper is much more readable, the technical contributions become more convincing, and the derivations are more clear. Most of my concerns have been addressed. I particularly like the revision of S2 in the supplementary document, as I found it much easier to follow now. I also think the current title more accurately reflects the contributions of this paper.

I don't quite agree with the text at Line 269. It sounds like it's straightforward to extend this method to other wavelength regimes. That's not necessarily true. Beside the fabrication and alignment difficulties (which the authors responded), some of the assumptions (such as the system being lossless) might not be easily held in other regimes. I suggest the authors discuss those potential challenges rather than just claiming "the method can be extended and applied in other wavelength regimes".

Other than that, I think the paper is in a good shape.

Changxi Zheng

Reviewer #2 (Remarks to the Author):

I am satisfied by the authors' responses, and am happy to recommend this work for publication in Nature Communications.

REVIEWERS' COMMENTS:

Reviewer #1 (Remarks to the Author):

Comments: Overall, I'm satisfied by this revision. The paper is much more readable, the technical contributions become more convincing, and the derivations are more clear. Most of my concerns have been addressed. I particularly like the revision of S2 in the supplementary document, as I found it much easier to follow now. I also think the current title more accurately reflects the contributions of this paper.

Response: We thank the referee for these positive feedbacks.

Comments: I don't quite agree with the text at Line 269. It sounds like it's straightforward to extend this method to other wavelength regimes. That's not necessarily true. Beside the fabrication and alignment difficulties (which the authors responded), some of the assumptions (such as the system being lossless) might not be easily held in other regimes. I suggest the authors discuss those potential challenges rather than just claiming "the method can be extended and applied in other wavelength regimes".

Other than that, I think the paper is in a good shape.

Response: We thank the referee for pointing out this severe shortcut, as indeed, downsizing the structure to nanoscale might not be that straightforward. Several limitations might result in complex design and fabrication, including the losses in metallic structures at optical frequencies and the need of high precision alignments of multilayers. We have modified the sentence and mild down the tone of this claim. The sentence reads now as follows.

Added or Revised Text:

To extend the method to shorter wavelengths, efforts in high precision lithography would be required. Moreover, metallic lossy plasmonic resonators to be used in replacement of current metallic patches would have to be replaced by chiral all-dielectric nano-antennas.

Reviewer #2 (Remarks to the Author):

Comments: I am satisfied by the authors' responses, and am happy to recommend this work for publication in Nature Communications

Response: We are grateful for the referee's positive feedback and recommendation.